# Preclinical and randomized clinical evaluation of the p38α kinase inhibitor neflamapimod for basal forebrain cholinergic degeneration

Ying Jiang[1,2,13], John J. Alam [3,13] ✉, Stephen N. Gomperts[4], Paul Maruff [5], Afina W. Lemstra[6,7], Ursula A. Germann[3], Philip H. Stavrides[1], Sandipkumar Darji[1], Sandeep Malampati[1], James Peddy[1], Cynthia Bleiwas[1], Monika Pawlik[1,2], Anna Pensalfini[1,2], Dun-Sheng Yang[1,2], Shivakumar Subbanna[1], Balapal S. Basavarajappa[1,2,8,9], John F. Smiley[1,2], Amanda Gardner[3], Kelly Blackburn[3], Hui-May Chu[10], Niels D. Prins[7], Charlotte E. Teunissen[6], John E. Harrison [6,11], Philip Scheltens[6] & Ralph A. Nixon[1,2,12] ✉

The endosome-associated GTPase Rab5 is a central player in the molecular mechanisms leading to degeneration of basal forebrain cholinergic neurons (BFCN), a long-standing target for drug development. As p38α is a Rab5 activator, we hypothesized that inhibition of this kinase holds potential as an approach to treat diseases associated with BFCN loss. Herein, we report that neflamapimod (oral small molecule p38α inhibitor) reduces Rab5 activity, reverses endosomal pathology, and restores the numbers and morphology of BFCNs in a mouse model that develops BFCN degeneration. We also report on the results of an exploratory (hypothesis-generating) phase 2a randomized double-blind 16-week placebo-controlled clinical trial (Clinical trial registration: NCT04001517/EudraCT #2019-001566-15) of neflamapimod in mild-to-moderate dementia with Lewy bodies (DLB), a disease in which BFCN degeneration is an important driver of disease expression. A total of 91 participants, all receiving background cholinesterase inhibitor therapy, were randomized 1:1 between neflamapimod 40 mg or matching placebo capsules (taken orally twice-daily if weight <80 kg or thrice-daily if weight >80 kg). Neflamapimod does not show an effect in the clinical study on the primary endpoint, a cognitive-test battery. On two secondary endpoints, a measure of functional mobility and a dementia rating-scale, improvements were seen that are consistent with an effect on BFCN function. Neflamapimod treatment is well-tolerated with no study drug associated treatment discontinuations. The combined preclinical and clinical observations inform on the validity of the Rab5-based pathogenic model of cholinergic degeneration and provide a foundation for confirmatory (hypothesis-testing) clinical evaluation of neflamapimod in DLB.

Degeneration of the basal forebrain, the primary source of cholinergic innervation in the brain, occurs in aging- and neurodegenerative disease-related cognitive disorders, including dementia with Lewy bodies (DLB), the second most common form of neurodegenerative dementia, where basal forebrain cholinergic neuron (BFCN) dysfunction leading to degeneration is considered to be an important driver of disease expression and progression[1-4]. In addition, BFCN degeneration has been proposed to be a major driver of the neurodegenerative process elsewhere, including in the hippocampus in Alzheimer's disease (AD)[5-7]. Importantly, recent studies have also demonstrated that BFCN loss underpins the gait dysfunction in Parkinson's disease (PD), suggesting that therapeutically targeting the cholinergic system could also address certain motor aspects of neurodegenerative diseases[8-10]. Recent evidence also indicates that correcting the functional deficit in the cholinergic system, thus amplifying physiologic release of acetylcholine, is likely to be more effective than the conventional approach of compensating for BFCN dysfunction by delaying clearance of released acetylcholine with cholinesterase inhibitors[11].

A critical pathogenic event in development of BFCN dysfunction and degeneration is impaired nerve growth factor (NGF) signaling, depriving cholinergic neurons of the neurotrophic support necessary for proper functioning and survival[12,13]. NGF signaling is transduced by endocytosis and retrograde trafficking of a maturing Rab5-"signaling endosome" containing the NGF receptor, TrkA, to initiate a transcriptional program. Cholinergic neurons, with long axonal projections throughout the cortex, are particularly vulnerable to disruption of this retrograde signaling process from distant synaptic connections back to the cell body. From a pathogenic mechanistic standpoint, the protein Rab5, a GTPase and master signaling molecule regulating endocytosis and endosome function, is implicated in development of impaired NGF signaling. Both Rab5 hyper-activation and lowered endosome recycling rates impair NGF signaling, partly due to enlargement of endosomes, which slows their retrograde transport and trophic signaling[14,15], leading to cholinergic atrophy in Down syndrome (DS) mouse models[13,14,16,17]. These DS models essentially recapitulate adult-onset basal forebrain cholinergic neurodegeneration[18]. Further, Rab5 hyperactivation in vivo in Rab5 overexpressing transgenic mice causes BFCN degeneration[18]. These findings establish Rab5 as a therapeutic target for BFCN dysfunction. Moreover, results from animal studies indicate that the neurodegeneration of BFCNs is reversible, as the BFCNs do not die with age and disease; rather, they lose cholinergic phenotype and functional properties, and this loss can be reversed by direct NGF infusion to the basal forebrain[19,20]. Thus, pharmacologically restoring NGF signaling, by targeting Rab5, has the potential to reverse disease progression by increasing numbers of functional cholinergic neurons.

As the alpha isoform of p38α kinase is a major Rab5 regulator and activator[21] and has been implicated in the regulation of retrograde axonal transport[22], we hypothesized that p38α inhibition would be a pharmacological approach to treating diseases associated with BFCN dysfunction. Accordingly, we evaluated a specific p38α kinase inhibitor, neflamapimod (NFMD)[23-25], preclinically in a transgenic DS mouse model (Ts2 mouse)[18] that demonstrates BFCN degeneration, as well as clinically in a placebo-controlled phase 2a study in patients with DLB[25], the primary objective of which was to evaluate the effects of neflamapimod on the cognitive domains impacted by the disease. When the clinical study was initiated, neflamapimod had been evaluated in phase 1 studies in healthy volunteers, in phase 2a studies in rheumatoid arthritis and AD[24], and in a then-ongoing 24-week 161-participant placebo-controlled phase 2 study in AD (since completed and reported, see Discussion)[25]. Herein we report on evidence of translation of the scientific understanding of mechanisms of cholinergic degeneration to effects on BFCNs both preclinically and clinically, with neflamapimod treatment (1) reversing pathological disease progression in the basal forebrain in the preclinical study in Ts2 mice; and (2) leading to outcomes consistent with a pharmacological effect on the basal forebrain cholinergic system in the clinical study of DLB.

## Results

### Neflamapimod treatment of Ts2 mice reverses endosomal pathology and restores number of cholinergic neurons

Ts2 mice, a DS mouse model that develops adult-onset Rab5+ endosomal pathology and cholinergic degeneration in the basal forebrain[18], and control 2 N (wildtype) mice were treated at approximately six months of age with either neflamapimod (3 mg/kg body weight) or vehicle (1% Pluronic F108), twice daily (BID) by oral gavage, for 28 days. The number and size of Rab5-GTP-positive endosomes, determined by ImageJ analysis, were greater in Ts2 than in wildtype mice while, with neflamapimod treatment, both parameters were normalized to values seen in the 2 N mice (Fig. 1a, b). Similarly, the intensity ratio of Rab5-GTP (activated form of Rab5) to total Rab5 was significantly higher in vehicle-treated Ts2 than in 2 N mice (Fig. 1c), was reduced to normal levels with neflamapimod treatment in Ts2 mice, and was not altered in 2 N mice. These results suggest that neflamapimod impacts aberrant, but not physiological, Rab5 activation. In Ts2 mice, basal forebrain neurodegeneration is well established by 6 to 7 months of age[18] and the number of medial septal choline acetyltransferase-positive (ChAT +) neurons (i.e., cholinergic neurons) continues to decline as the mice age (Fig. 1d). Neflamapimod-treated Ts2 mice had significantly higher numbers of ChAT+ neurons at the end of the four-week treatment period than did vehicle-treated Ts2 mice ($p = 0.0037$), exhibiting numbers comparable to those in the 2 N mice (Fig. 1e, f). Notably, in vehicle-treated Ts2 mice, ChAT+ neurons had an abnormal morphology, with neurite swelling (yellow arrows) or atrophy (orange arrow), perikaryal atrophy (red arrows), and below normal ChAT immunoreactivity (green arrows) (Fig. 1h), while neflamapimod treatment appeared to normalize their morphology (Fig. 1g, h).

Synaptic function, as represented by the long-term potentiation (LTP) of synaptic transmission in the Schaffer collateral synapses (CA3-CA1) of Ts2 mice, was improved modestly ($p = 0.0334$, 18% increase in fEPSP slope at 110 minutes) by neflamapimod treatment (Fig. 2a–c). Behavioral evaluations considered dependent on basal forebrain cholinergic function, novel object recognition (NOR) and open field tests, were examined in a separate cohort of animals. Because of limited availability of Ts2 mice at the time, no vehicle-administered animals could be included in these evaluations. Instead, the behavioral tests were performed both pre- and post- four weeks dosing with neflamapimod and, in addition, 2 N mice were included for comparison. For both the NOR and open field tests, there was a significant difference between 2 N and Ts2 mice at pretreatment that was not apparent at the end of neflamapimod treatment (Fig. 2d, e). For the open field test, performance at the end of treatment was significantly improved from that at the pretreatment baseline (Fig. 2e).

### The positive effects of neflamapimod on pathology and function in Ts2 mice appear to be mediated by p38α kinase inhibition

Further biochemical analyses were conducted to understand the underlying mechanisms. First, p38, phosphorylated-p38 (p-p38) and its downstream substrates MK2[26] and MNK1[26,27] were assessed in tissue homogenates of the brain cortex by western blotting, as described previously[18]. Indicating target engagement, neflamapimod treatment reduced p-p38 levels, while also significantly decreasing levels of total MK2 and MNK1, along with pMNK1, in Ts2 mice (Fig. 3a, b).

As reduction of p38α activity in APP/PS1 double transgenic mice leads to decreased BACE1 protein expression[28], BACE1 protein levels in the cortex were similarly evaluated. Indeed, Ts2 mice treated with neflamapimod, compared with vehicle, had significantly lower BACE1 protein levels (Fig. 3c, d; $p = 0.047$ for neflamapimod vs. vehicle-

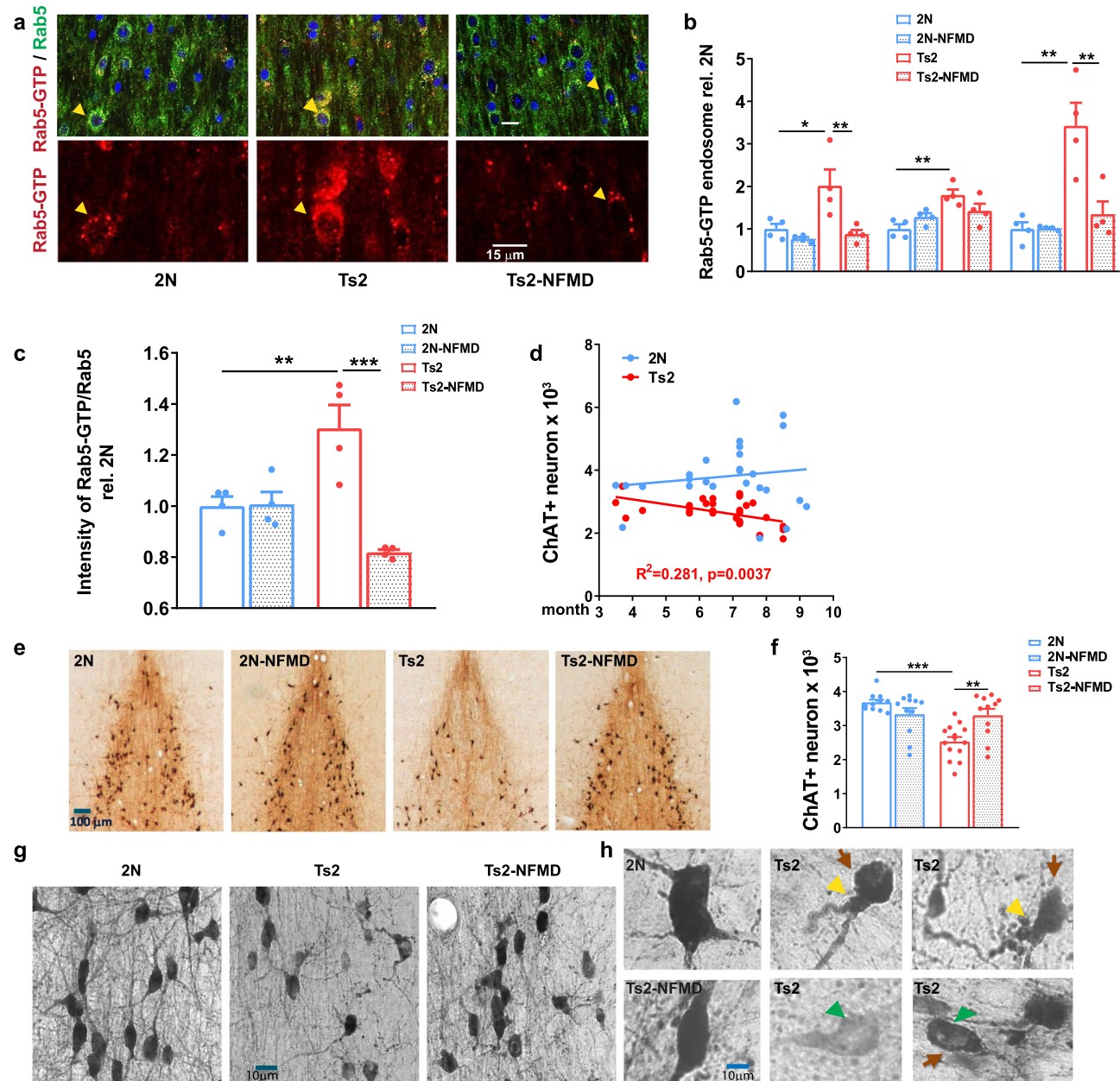

**Fig. 1 | Normalized Rab5+ endosome number/size and restored cholinergic neuronal numbers in Ts2 mice treated with neflamapimod (NFMD).**
**a** Representative images of medial septal nucleus (MSN) regions from vehicle-treated wildtype (2 N, $n = 4$), vehicle-treated Ts2 ($n = 4$) and NFMD-treated Ts2 (Ts2-NFMD) mice, labeled with Rab5-GTP (red) and Rab5a (green) antibodies; arrows point to the Rab5-positive neurons, which are shown enlarged in the lower panel (scale bar, 15 μm). **b** The numbers, sizes and areas of Rab5-GTP-positive endosomes determined by Image J analysis [$n = 4$ mice per group; for number: $F_{(3,12)} = 7.789$, R square = 0.661; for 2 N vs Ts2 $p = 0.0202$, 95% CI = −1.880, −0.151; for Ts2 vs Ts2-NFMD $p = 0.0097$, 95% CI = 0.273, 2.002; for size: $F_{(3,12)} = 6.939$, R square = 0.634, for 2 N vs Ts2 $p = 0.0036$, 95% CI = −1.329, −0.269; for area: $F_{(3,12)} = 13.19$, R square = 0.767; for 2 N vs Ts2 $p = 0.0009$; 95% CI = −3.766, −1.079; for Ts2 vs Ts2-NFMD $p = 0.0030$, 95% CI = 0.7325, 3.419]. **c** The ratio of Rab5-GTP to total Rab5 intensities determined with Image J [$n = 4$ mice per group; $F_{(3,12)} = 13.23$, R square = 0.768; for 2 N vs Ts2 $p = 0.001$, 95% CI = −0.537, −0.073; for Ts2 vs Ts2-NFMD $p = 0.0002$, 95% CI = 0.254, 0.719]. **d** The number of stereologically counted

ChAT+ neurons in the MSN region of 2 N ($n = 27$) and Ts2 ($n = 28$) mice was graphed versus age of mice in months (Linear regression F = 10.16, R square = 0.281, $p = 0.0037$, 95% CI = −0.259, −0.056 for Ts2; F = 0.548, R square = 0.0215, $p = 0.466$, 95% CI = -infinity, −3.98) for 2N. **e** Representative images of diaminobenzidine (DAB)-stained ChAT+ neurons in the MSN region of 2 N and Ts2 mice treated with either vehicle or NFMD (scale bar, 100 μm). **f** Quantification of DAB-stained ChAT+ neurons in the MSN region; $n = 10$ (2 N), $n = 11$ (2N-NFMD), $n = 14$ (Ts2) and $n = 11$ (Ts2-NFMD) mice ($F_{(3, 43)} = 11.10$, R square = 0.436, for 2 N vs Ts2, $p < 0.0001$, 95% CI = 0.588, 1.712; for Ts2 vs Ts2-NFMD $p = 0.0037$, 95% CI = −1.331, −207).
**g**, **h** Abnormal morphology of ChAT+ neurons was consistently and reproducibly seen in the MSN of Ts2 mice, with swelling (yellow arrows), perikaryal atrophy (red arrows) and below normal ChAT immunoreactivity intensity (green arrows) in the representative images (scale bar,10 μm). Data are presented as mean values ± SEM. Graph made and analyzed with GraphPad Prism8.0.1 with Ordinary One-Way ANOVA/Tukey correction. Statistical significance is represented by asterisks $*p \leq 0.05$, $**p \leq 0.01$, $***p \leq 0.001$. Source data are provided as a Source Data File.

treated Ts2 mice). In addition, levels of βCTF, the product of APP cleavage by BACE1, which were significantly higher in Ts2 than in wildtype mice ($p = 0.0003$), were decreased significantly by neflamapimod treatment in the Ts2 mice (Fig. 3c, d, $p = 0.0007$).

## Effect of neflamapimod on the primary outcome measure in an exploratory phase 2a clinical study in DLB

The AscenD-LB study (NCT04001517) was a 91-patient, 16-week, double-blind placebo-controlled phase 2a clinical study in patients

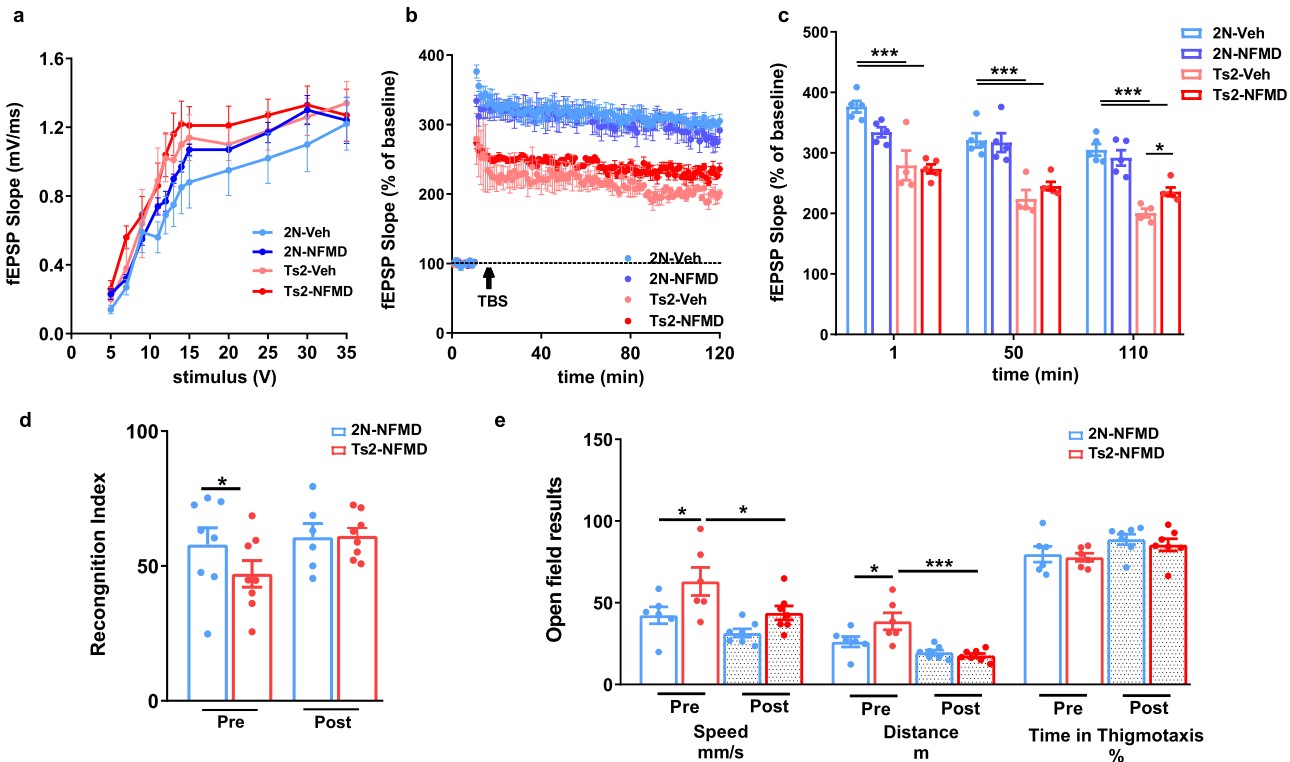

**Fig. 2 | Slower decline of LTP and normalized scores of behavioral tests in Ts2 mice treated with NFMD. a** Input/output relationship plots of hippocampal slices from 2 N and Ts2 mice treated with either vehicle or NFMD (*n* = 4 mice for Ts2 group and *n* = 5 mice for the other treatment groups), and graphed using GraphPad Prism8.0.1 [no significant difference between the slopes, Linear regression F (3, 40) = 0.107, *p* = 0.956]. **b** Plots of LTP in the Schaffer collateral synapses (CA3-CA1) induced by theta-burst stimulation (TBS) of mice from four treatment groups [Linear regression for, significant difference between the slopes of 2 N vs Ts2, F(1, 236) = 6.925, *p* = 0.0091; no significant difference between the slopes of Ts2 vs Ts2-NFMD, F(1, 236) = 2.103, *p* = 0.148]. **c** Averages of fEPSP slopes at 1, 50, and 110 min following tetanic stimulation; NFMD treatment increased the fEPSP slope of Ts2 mice by 18% at the 110 min time point [Ordinary One-Way ANOVA with Tukey correction, at 1 min time point, F(3, 34) = 20.75, R square = 0.647, for 2 N vs Ts2, *p* < 0.0001, 95% CI = 54.62, 139.5; at 50 min time point, F(3, 34) = 16.44, R square = 0.592, for 2 N vs Ts2, *p* < 0.0001, 95% CI = 49.54, 144.5; at 110 min time point, F(3, 34) = 19.00, R square = 0.626, for 2 N vs Ts2, *p* < 0.0001, 95% CI = 72.09, 139.6, for

Ts2 vs Ts2-NFMD, *p* = 0.0334, 95% CI = −67.71, −2.938]. **d** Novel Object Recognition (NOR) test at 24 h after familiarization session, represented by recognition index in 2 N (*n* = 8) and Ts2 (*n* = 8) mice before (Pre) and after (Post) 4 weeks of NFMD treatments [Ordinary One-Way ANOVA with Tukey correction, F(3, 24) = 1.967, R square = 0.197, for Ts2-pre vs Ts2-post, *p* = 0.0262, 95% CI = 2.406, 34.97]. **e** Open field test results including speed, distance and percentage of time spent in thigmotaxis for 2 N (*n* = 8) and Ts2 mice (*n* = 8) Pre and Post 4 weeks of NFMD treatment [Ordinary One-Way ANOVA with Tukey correction, For Speed F(3, 22) = 5.936, R square = 0.447, for 2N-NFMD-pre vs Ts2-NFMD-pre, *p* = 0.0148, 95% CI = −36.97, −4.47; for Ts2-NFMD-pre vs Ts2-NFMD-post *p* = 0.0148, 95% CI = 3.572, 34.89; For Distance F(3, 22) = 9.784, R square = 0.572, for 2N-NFMD-pre vs Ts2-NFMD-pre, *p* = 0.0434, 95% CI = −27.75, −0.295; for Ts2-NFMD-pre vs Ts2-NFMD-post *p* = 0.0003, 95% CI = 9.205, 32.77). Data are presented as mean values ± SEM. Statistical significance represented by asterisks *$p \le 0.05$, **$p \le 0.01$, ***$p \le 0.001$. Source data are provided as a Source Data File.

with mild-to-moderate probable DLB [dementia, with at least one core clinical feature of DLB and demonstrated abnormality in dopamine uptake by DaTscan™ (Ioflupane I123 SPECT), consistent with 2017 consensus clinical criteria[29,30]; see also Methods] and receiving cholinesterase inhibitor therapy (>3 months, stable dose >6 weeks). The main objective of this phase 2a study was to evaluate the effects of neflamapimod on the cognitive domains that are impacted in DLB. Accordingly, the primary outcome measure was a study-specific six-test neuropsychological test battery (NTB) that assessed attention (Identification, Detection tests), executive function (Category Fluency, Letter Fluency, One Back accuracy), and visual learning (One Card Learning). Secondary objectives included evaluation of neflamapimod effects on a dementia rating scale, episodic memory, motor function, and neuropsychiatric outcomes. As there was not a pre-specified hypothesis for treatment effect relative to placebo, no formal sample size calculations were performed but, based on prior experience with use of similar cognitive test battery in a prior DLB clinical trial[31], 40 participants per arm was considered sufficient to provide a first assessment of the ability of neflamapimod to improve cognitive function in patients with DLB.

Patients were randomized 1:1 to either neflamapimod 40 mg capsules or matching placebo and then, based on body weight, assigned to either a twice-daily (BID) [weight <80 kg; 40 mg BID neflamapimod or placebo BID] or thrice-daily (TID) [weight ≥80 kg; 40 mg TID neflamapimod or placebo TID] regimen (see Consort Flow Diagram, Fig. 4). This dosing regimen was utilized because, based on available preclinical and clinical data at the time the study was initiated, the target for therapeutic efficacy was a 12 h plasma drug exposure of 100 ng × h/mL; and the pharmacokinetic data available at this time predicted that 40 mg BID would achieve that target, but only in patients weighing less than 80 kg (thus requiring 40 mg TID in the higher weight range). In addition, prior clinical data suggested that there was an effect of weight on clearance and that a dose of 40 mg TID in patients weighing <80 kg would exceed a limit imposed at the time by a regulatory agency to provide a ten-fold safety margin to the no-adverse-effect-level in long-term animal toxicity studies.

All efficacy endpoints were analyzed as change from baseline using a linear mixed model for repeated measures (MMRM), with baseline as a covariate, over the entire course of the study (i.e., all time points were included in the analysis). Consistent with the objective of

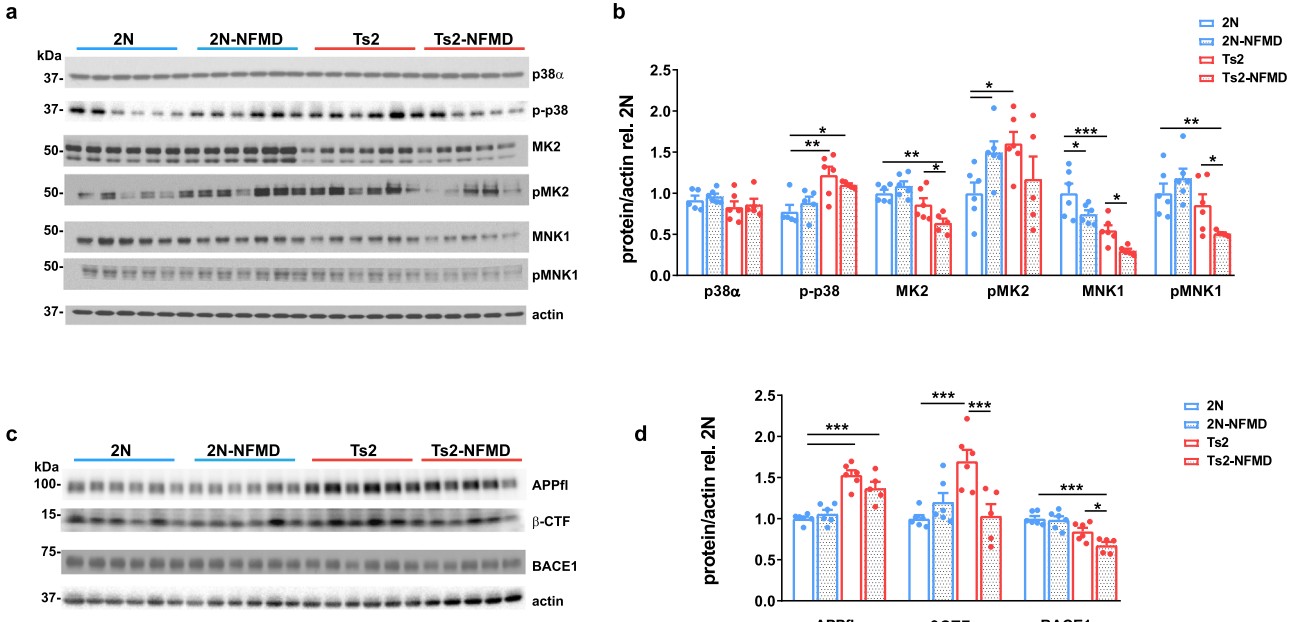

**Fig. 3 | The positive effects of neflamapimod on pathology and function in Ts2 mice appear to be mediated by inhibition of the p38α kinase pathway and related to BACE1 and βCTF reductions. a** Western blot analysis of p38α, phosphorylated-p38 (p-p38), and downstream substrates MK2 and MNK1 in tissue homogenates of the brain cortex after 2 weeks of either vehicle or NFMD treatment ($n = 5$ for Ts2-NFMD, and $n = 6$ for other treatment groups). **b** Quantification of the western blot images shown with Image J and graphed with GraphPad Prism 8.0.1 [Ordinary One-Way ANOVA, For p-p38 $F_{(3, 19)} = 6.658$, R square = 0.540, for 2 N vs Ts2, $p = 0.0044$, 95% CI = −0.760, −0.131; for 2 N vs Ts2-NFMD $p = 0.0495$, 95% CI = −0.656 −0.000194; For MK2 $F_{(3, 19)} = 10.33$, R square = 0.610, for 2 N vs Ts2-NFMD, $p = 0.0005$, 95% CI = 0.182, 0.541; for Ts2 vs Ts2-NFMD $p = 0.0166$, 95% CI = 0.0457, 0.404; For pMK2 $F_{(3, 19)} = 2.850$, R square = 0.310, for 2 N vs Ts2, $p = 0.0178$, 95% CI = −1.095, −0.117; for 2 N vs 2N-NFMD $p = 0.0469$, 95% CI = −0.985, −0.0076; for Ts2 vs Ts2-NFMD $p = 0.0166$, 95% CI = 0.0457, 0.404; For MNK1 $F_{(3, 19)} = 15.11$, R square = 0.705, for 2 N vs Ts2, $p = 0.0003$, 95% CI = 0.235, 0.668; for 2 N

vs 2N-NFMD $p = 0.023$, 95% CI = 0.0393, 0.472, for Ts2 vs Ts2-NFMD, $p = 0.0347$, 95% CI = 0.0197, 0.473; For pMNK1 $F_{(3, 19)} = 6.285$, R square = 0.498, for 2 N vs Ts2-NFMD, $p = 0.0063$, 95% CI = 0.156, 0.824; for Ts2 vs Ts2-NFMD $p = 0.0428$, 95% CI = 0.125, 0.681]. **c, d** Subsequent western blot analysis for full length APP (APPfl), APP-βCTF and BACE1 and western blot image quantified with Image J and graphed with GraphPad Prism 8.0.1 (Ordinary One-Way ANOVA, For APPfl $F_{(3, 19)} = 23.33$, R square = 0.786, for 2 N vs Ts2, $p < 0.0001$, 95% CI = −684, −0.376; for 2 N vs Ts2-NFMD $p = 0.0001$, 95% CI = −0.532, −0.209; For APP-βCTF $F_{(3, 19)} = 8.167$, R square = 0.563, for 2 N vs Ts2, $p = 0.0003$, 95% CI = −1.027, −0.367; for Ts2 vs Ts2-NFMD $p = 0.0007$, 95% CI = 0.3187, 1.010; For BACE1 $F_{(3, 19)} = 12.86$, R square = 0.670, for 2 N vs Ts2-NFMD, $p = 0.0001$, 95% CI = 0.158, 0.491; for Ts2 vs TS2-NFMD $p = 0.0469$, 95% CI = −0.00185, 0.335). Data are presented as mean values ± SEM. Statistical significance is represented by asterisks *$p \le 0.05$, **$p \le 0.01$, ***$p \le 0.001$. Source data are provided as a Source Data File.

---

evaluating treatment effects relative to placebo, rather than testing hypotheses that treatment was superior to placebo, differences between neflamapimod and placebo, with 95% confidence intervals (CI), from the MMRM analysis are reported. Also reported are $p$-values for the protocol-specified analyses, though not for the purposes of significance (i.e., hypothesis) testing, rather for the purposes of evaluating the strength of evidence for a treatment effect. As this was an exploratory study and primary and secondary endpoints are independent of each other, $p$-values were not adjusted for multiplicity. The Efficacy Population used for efficacy analyses, consistent with the intent-to-treat (ITT) principles included all subjects who have a baseline and at least one on-study assessment of the efficacy parameter.

There were no significant differences in baseline demographic and disease characteristics among groups, either comparing all neflamapimod recipients *vs.* placebo, or 40 mg TID vs. either placebo or placebo TID (Table 1).

As the six tests in the NTB have different units, the results of the individual tests at each time point were normalized by converting to z-scores and then combined into a single z-score in which the individual tests were weighted equally. MMRM analysis revealed no to minimal treatment effect on this primary outcome measure, the mean NTB composite z-score, (Table 2; $p > 0.2$, drug-placebo difference = 0.04, 95% CI:−0.11, 0.19; Cohen's d effect size for improvement, $d = 0.10$). As pre-specified in the statistical analysis plan, the two tests within the NTB that measure attention and processing speed (i.e., the Detection and Identification Tests) were analyzed as an Attention

Composite. A small potential benefit for neflamapimod over placebo was observed in z-score of this Attention Composite (Table 2, $p = 0.17$, drug-placebo difference = 0.14, 95% CI: −0.06, 0.36; $d = .18$). In the placebo group, both cognitive composites remained stable or worsened slightly over the 16 weeks of the study.

### Effects of neflamapimod on the secondary outcome measures in the clinical study

The Clinical Dementia Rating Scale Sum of Boxes (CDR-SB) scale, designed to assess both cognition and function, scores six domains (memory, orientation, judgment & problem solving, community affairs, home & hobbies, and personal care) on a 0–3 scale (total range 0–18, higher scores indicating worse dementia). The CDR-SB was performed at baseline, week 8, and week 16. Comparing all neflamapimod with all placebo, there was improvement with neflamapimod treatment (Table 2, $p = 0.023$, drug-placebo difference = −0.45, 95% CI:−0.83,−0.06, $d = 0.31$).

Neflamapimod treatment did not impact the cognitive test in the study considered dependent on hippocampal function, a verbal learning test (International Shopping List Test, ISLT; Table 2, $p > 0.2$, drug-placebo difference = 0.17, 95%CI −1.61, 0.87, $d = −0.02$) included to evaluate effects of neflamapimod on episodic memory. The final cognitive test evaluated in the study was the Mini Mental Status Examination (MMSE), a measure of global cognition and, as such, not specifically linked to either BFCN or hippocampal function. At baseline, mean MMSE scores were high (Table 1) for a mild-to-moderate

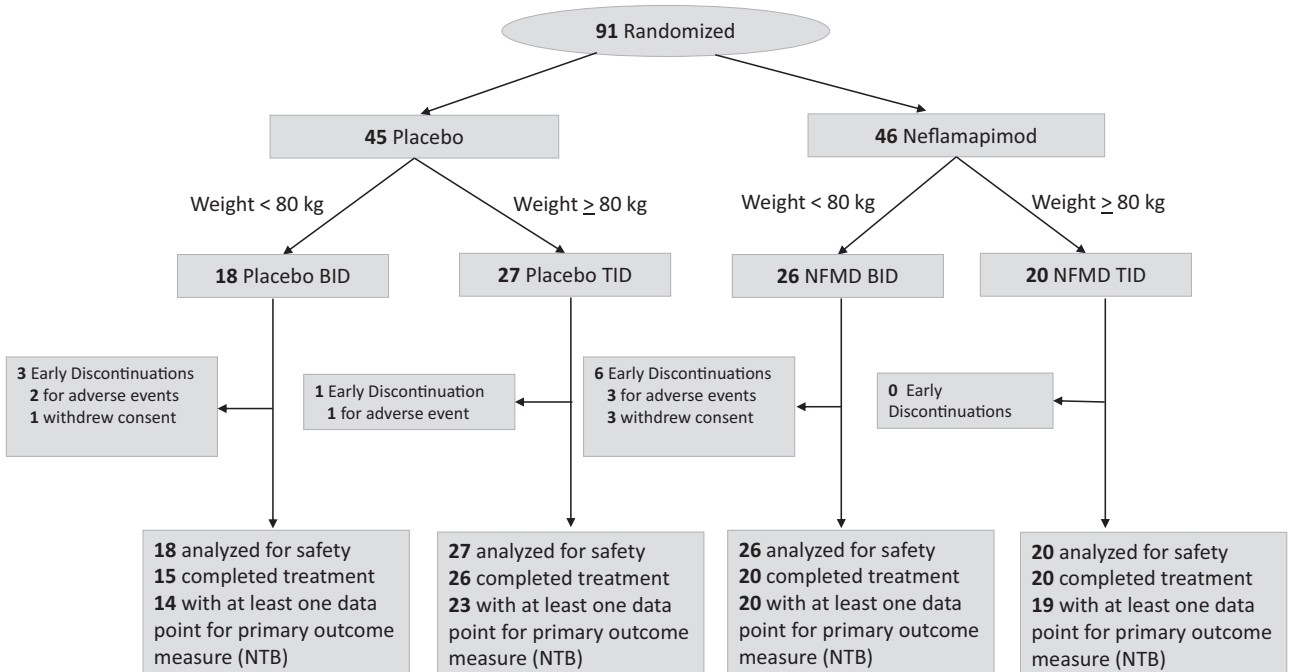

**Fig. 4 | CONSORT flow diagram for the AscenD-LB Phase 2a clinical trial.** CONSORT 2010 Statement flow diagram showing participant flow through each stage of the AscenD-LB phase 2a randomized controlled trial from randomization through to follow-up and data analysis.

dementia population, reflecting the insensitivity of the MMSE in the early stages of DLB such as the one in the current study[32]. Interpretation of the effects on the MMSE were also limited by COVID-19 pandemic restrictions, which led to a third of on-study evaluations either being either missed or conducted remotely via video, an approach that has not been validated. As such, MMRM analysis was not performed for this outcome measure. Within each category of assessment, onsite or remote, there were no differences among treatment groups on the MMSE (Suppl. Table 1).

Gait dysfunction is very common in patients with DLB and recent evidence indicates that gait dysfunction in that context is intimately related to atrophy of the nucleus basalis of Meynert within the basal forebrain cholinergic system[8]. Gait in the AscenD-LB study was assessed with the Timed Up and Go (TUG) test measuring functional mobility by monitoring the time that a subject takes to rise from a chair, walk three meters, turn around 180 degrees, walk back to the chair, and sit down while turning 180 degrees. In the comparison of the combined neflamapimod groups to placebo, there was reduction in the time required to complete the TUG test (i.e., improvement) with neflamapimod treatment (Table 2, $p = 0.044$, drug-placebo difference = −1.4 sec, 95%CI:−0.1 to −2.7, $d = 0.22$).

### Neuropsychiatric outcomes and safety in the clinical study
Four specific domains within the 10-item Neuropsychiatric Inventory [depression (dysphoria), anxiety, hallucinations, and agitation/aggression] were secondary outcome measures in the study. Of these, the domain most specific for both DLB and the cholinergic system is hallucinations[33], for which there was a difference favouring neflamapimod for severity (Suppl. Fig. 1; $p = 0.051$, drug-placebo difference = −0.29, 95%CI:−0.43, −0.01, for all neflamapimod vs. placebo). There were no discernible treatment effects for the other three domains of the NPI-10 (Supp. Fig. 2).

Neflamapimod treatment was well-tolerated with no study drug associated treatment discontinuations or serious adverse events (SAEs) reported (See Suppl. Table 2 for all treatment discontinuations and SAEs). Treatment-emergent adverse events occurring at an incidence >5%, in either neflamapimod- or placebo-treated patients, were

falls (13% in neflamapimod; 9% in placebo), headache (9%, 4%), diarrhea (7%, 11%), nausea (7%, 7%), and tremor (0%, 7%). There was one reported event of worsening of parkinsonism, reported in a participant receiving placebo. Among neflamapimod-treated patients, incidences of diarrhea and headache (15% each) were higher in those receiving 40 mg TID than in those receiving 40 mg BID (0% diarrhea; 3% headache); the incidence of falls was higher in those receiving 40 mg BID (19%) than in those receiving 40 mg TID (5%).

### Exploratory analyses of the clinical study to evaluate the effects of dose on efficacy outcomes
During the conduct of the AscenD-LB study, results from the 161-patient study in AD[25] became available and these showed that the effect of weight on plasma neflamapimod drug concentration levels was limited to patients weighing <60 kg and otherwise the mean 12 h plasma drug exposure with 40 mg BID, the only dose utilized in that study, was only 70 ng*hr/mL (i.e. 30% lower than targeted for the current study). As only one neflamapimod participant in the current study weighed <60 kg, and as 40 mg BID was inefficacious in the AD study, the statistical analysis plan (SAP) for the AscenD-LB study was written allow for the conduct, in parallel with the protocol-defined analyses, of additional MMRM analyses of the efficacy outcome measures to compare the dose group that achieved targeted plasma drug concentrations (i.e., 40 mg TID) to placebo. Importantly, in the current study the measured plasma drug concentrations, available after the statistical analyses were completed, were 50% higher in 40 mg TID recipients, compared to 40 mg BID (median steady-state concentration of 6.8 ng/mL in 40 mg BID vs 10.2 ng/mL in 40 mg TID). In these additional analyses (Suppl. Table 3, Suppl. Figs. 3 and 4) for both the NTB Composite and the Attention Composite, the z-score in 40 mg TID participants improved from baseline and was different over the course of the study from placebo (NTB Composite drug-placebo difference = 0.18 95% CI:0.00-0.35, $d = 0.47$; Attention Composite drug-placebo difference = 0.28, 95% CI 0.04−0.52, $d = 0.41$). Improvement relative to placebo was also seen with 40 mg TID in the CDR-SB (drug-placebo difference = −0.56, 95% CI:−0.96, −0.16, $d = 0.35$) and the TUG Test (drug-placebo difference = −1.4 sec, 95% CI:−0.2,−2.6, $d = 0.50$). To account for potential effects of

**Table 1 | Baseline characteristics in the clinical study**

|  | Placebo (n = 45) | NFMD ALL (n = 46) | Placebo TID (n = 27) | NFMD 40 mg TID (n = 20) |
|---|---|---|---|---|
| Age (yrs) | 72.1 (6.9) | 73.5 (6.9) | 70.4 (5.7) | 72.2 (6.6) |
| Age range | 62–87 | 59–85 | 62–83 | 59–84 |
| Male | 87% | 85% | 96% | 95% |
| CDR Sum of Boxes | 5.1 (3.2) | 4.9 (1.8) | 4.4 (2.3) | 4.7 (1.8) |
| MMSE | 23.0 (3.3) | 23.1 (3.9) | 23.6 (3.3) | 23.4 (3.3) |
| ISLT | 14.3 (5.4) | 13.6 (5.9) | 14.2 (6.2) | 14.1 (4.9) |
| Timed Up and Go (seconds) | 13.5 (6.4) | 12.7 (3.7) | 13.3 (5.2) | 13.3 (3.8) |
| Fluctuating cognition | 60% | 61% | 55% | 60% |
| Visual hallucinations | 55% | 61% | 48% | 70% |
| REM sleep disorder | 73% | 59% | 78% | 70% |
| Parkinsonism | 84% | 78% | 78% | 75% |
| ≥2 core clinical features* | 87% | 83% | 82% | 90% |
| Carbidopa-Levodopa | 29% | 28% | 37% | 40% |
| Anti-depressants | 49% | 54% | 52% | 65% |
| Clonazepam | 15% | 15% | 26% | 30% |
| Baseline plasma ptau181<2.2 pg/mL | 51% | 55% | 54% | 55% |

*Fluctuating cognition, visual hallucinations, REM sleep disorder, or parkinsonism.

**Table 2 | Efficacy outcome measures in the clinical study**

|  | All Neflamapimod (NFMD; includes 40 mg BID and 40 mg TID participants) vs. All Placebo | | | | | |
|---|---|---|---|---|---|---|
| Outcome measure | Number of particpants | | Mean baseline values | | Change from baseline | |
|  | NFMD | Placebo | NFMD | Placebo | Drug-Placebo Difference On-Study (95% CI) | p-value | Cohen's d Effect Size for Improvement - d |
| NTB* Composite | 39 | 37 | 0.04 | 0.05 | 0.04 (−0.11, 0.19) | >0.2 | 0.10 |
| Attention Composite | 39 | 36 | 0.04 | −0.02 | 0.14 (−0.06, 0.35) | 0.17 | 0.18 |
| Clinical Dementia Rating Sum of Boxes (CDR-SB) | 41 | 42 | 4.9 | 5.1 | −0.45 (−0.83, −0.06) | 0.023 | 0.31 |
| International Shopping List Test (ISLT) | 42 | 42 | 14.3 | 13.6 | −0.17 (−1.61, 0.87) | >0.2 | −0.02 |
| Timed Up and Go (TUG) | 39 | 38 | 12.7 | 13.5 | −1.4 (−2.7, −0.1) | 0.044 | 0.22 |

*NTB: Neuropsychological Test Battery evaluating attention, executive function, and visual learning.
The NTB was the primary outcome measure. NTB and Attention composites reported as z-scores.
Note: Difference (95% confidence interval, CI) shown is from MMRM (mixed model for repeated measures) analysis. Improvement is reflected as increases in NTB, Attention Composite and the ISLT; and as decreases in CDR-SB and TUG test. Positive d indicates improvement relative to placebo, and negative d indicates worsening relative to placebo.

weight, MMRM analyses were also conducted for the comparison of the two higher weight cohorts, neflamapimod 40 mg TID and placebo TID (Suppl. Table 4); the results of which were similar to that seen in comparison of 40 mg TID and placebo.

## Discussion

Consistent with our overarching hypothesis, Ts2 (DS) mice with established basal forebrain pathology, when treated with the p38α inhibitor neflamapimod, showed decreased Rab5 activation, reversal of Rab5+ endosomal pathology, and, most important, restoration of the number and morphology of cholinergic neurons in the medial septal region of the basal forebrain. As with NGF studies, we interpret the increased cholinergic neuron numbers as restoration of the cholinergic phenotype in neurons that have, otherwise, continued to survive, rather than as a neuro-regenerative effect[19,34]. Supporting that interpretation, p38 MAPK has been implicated in cholinergic differentiation[35]. Moreover, inflammation, in a p38 MAPK-dependent manner, reversibly reduced ChAT expression in the basal forebrain[36]. Though not a main aim of the Ts2 mouse study, behavioral assessments did indicate a functional impact of increasing the number of cholinergic neurons, with neflamapimod treatment restoring performance to wildtype levels in behavioral tests linked to cholinergic function, the open field and NOR tests[37–39]. In contrast, neflamapimod treatment had only minor effects on synaptic plasticity in the hippocampus, where the cholinergic system may not be the major determinant of the outcome, as defects in hippocampal function in the DS mouse model may be independent of cholinergic transmission[40] and represent fixed structural deficits, including frank neuronal loss[41]. Our functional results may also be evaluated in the context of published studies with neflamapimod in rat models, where behavioral outcomes are more readily assessed[23]. First, neflamapimod reversed deficits in Morris water maze performance[23], attributed to basal forebrain dysfunction[42], in aged rats. Second, neflamapimod improved functional recovery after a transient-ischemia induced stroke in rats[43], with basal forebrain cholinergic function regarded as rate-limiting for the recovery process[44]. Together with these previously reported findings, the effects of neflamapimod in Ts2 mice provide preclinical proof-of-concept for p38α inhibition to reverse BFCN dysfunction.

The preclinical mechanistic findings were consistent with our original hypothesis that p38α inhibition is an approach to decrease Rab5

activation and reverse the block in endosomal trafficking and signaling. Specifically, neflamapimod treatment decreased Rab5-GTP levels, a marker of Rab5 activation, and reversed the Rab5+endosomal pathology that is a direct histopathological marker of the block in endosomal trafficking underlying the defect in NGF signaling in Ts2 mice[12,14]. In separate experiments, restoration of cholinergic neuron numbers was observed following neflamapimod treatment in Rab5-overexpressing transgenic mice, which otherwise develop basal forebrain cholinergic degeneration[45]; this further supported the activity of neflamapimod on Rab5-mediated cholinergic neuron loss. The beneficial effects were associated with p38α inhibition, because neflamapimod treatment normalized levels of activated, i.e., phosphorylated p38α (in addition to being activated by upstream kinases, p38α auto-activates[46]) and lowered levels of its downstream substrates MK2 and MNK1. Furthermore, neflamapimod treatment decreased BACE1 protein levels, consistent with reports that neuronal p38α knockout in APP/PS1 ("Alzheimer's") transgenic mice led to decreased BACE1 protein levels, through increased autophagy-lysosome mediated degradation of the BACE1 protein[28,47,48]. Moreover, in a recent study, p38α activity decreased BACE1 levels in synaptic terminals by increasing retrograde axonal transport of BACE1 to lysosomes for degradation[49]. Thus, the finding that neflamapimod decreases BACE1 protein levels in Ts2 mice provides further evidence that the drug improves retrograde axonal transport of endosomes, the fundamental pathophysiologic defect being targeted with our therapeutic approach. Finally, the reduction in βCTF levels, resulting from decreased BACE1 protein levels, is also a likely contributor to the effects on cholinergic degeneration and function[14,18].

With a mechanistic foundation provided by the animal studies, the results of the clinical study in DLB, a neurodegenerative disease characterized by substantial basal forebrain cholinergic loss provides an opportunity to understand whether the preclinical results with respect to BFCN function could be translated to the clinic. While there was no apparent treatment effect on the primary outcome measure, an NTB designed to evaluate effects on the cognitive deficits most prominent in DLB, when interpreted in the context of the recent advances in the understanding of the network connections and function of the basal forebrain cholinergic system as well mechanistic insights from the preclinical study, a number of the findings in the neflamapimod clinical study reflect outcomes that would be predicted for a drug acting pharmacologically against BFCN dysfunction. For example, the tests within the NTB most responsive to treatment with neflamapimod measure visual attention, while in a recent report mapping changes in cholinergic networks in DLB using PET imaging with an acetylcholinesterase transporter ligand, functional networks related to alertness and visual attention were most prominently disrupted[50]. In addition basal forebrain pathology (volume by MRI) has been linked to attention and attentional deficits[51,52]. The results in the TUG test, showing gait improvement relative to placebo, may be particularly relevant with regard to a potential effect on the cholinergic system because recent clinical translational publications have connected basal forebrain pathology (assessed as NBM atrophy by MRI) to gait impairment in PD[8-10]. That is, gait dysfunction is a clinical correlate and potentially clinical biomarker of basal forebrain pathology. In addition, in a recent study in patients with DLB or PD undergoing deep brain stimulation of the NBM, a prominent functional network from the NBM to the supplemental motor cortex was defined and proposed to be involved in volitional control of movement[53]. Thus, the various translational studies in the literature suggest that the apparent treatment effects on the TUG test may be providing direct support for activity of neflamapimod against pathologic BFCN dysfunction, akin to its effects on BFCN pathology in the preclinical study; evidence that is independent of effects on cognition. Moreover, as cholinesterase inhibitors can improve cognition but generally provide no benefit to motor function[54], one hypothesis for the relative lack of effects on the cognitive endpoints (relative to that seen in the TUG) is that the cholinesterase inhibitors had provided much of the cognitive benefit possible in the study, which limited the ability to demonstrate a cognitive effect, but did not impact the ability to demonstrate motor function effects on the TUG test. The same theoretical argument would apply the CDR-SB, which also has the ability to demonstrate motor function effects through the functional domains (e.g., personal care). Consistent with that notion, the most prominent treatment effect relative to placebo within the individual domains of the CDR-SB was on the personal care domain (Suppl. Fig. 5). Otherwise, as basal forebrain pathology (volume of Nucleus basalis of Meynert by MRI) is strongly linked to the dementia (i.e., cognitive dysfunction) in DLB[55], in this context, and not necessarily in other dementias, treatment effects on the basal forebrain would be expected to be reflected in the CDR-SB.

To further explore the translational science, plasma levels of a biomarker for the presence of AD co-pathology (tau phosphorylated at position 181, ptau181[56]) were measured in baseline samples and the results evaluated by the presence or absence of AD co-pathology (i.e., plasma ptau181 < or ≥ 2.2 pg/mL). This analysis tested the concept that the most pure evaluation of a drug acting on BFCN function would be to assess treatment effects in patients with DLB in whom BFCN pathology is predominant over AD pathology, which is represented by patients who do not have AD co-pathology (patients with DLB, otherwise, consistently have basal forebrain cholinergic degeneration[57]). Indeed, in those patients within the current study who do not exhibit AD co-pathology (i.e., those with baseline plasma ptau181 < 2.2 ng/mL) the magnitude of the treatment effect was particularly high and greater than in patients with mixed DLB-AD pathology (Suppl. Fig. 6). These findings provide additional support both for a pharmacological effect on the basal forebrain and for that effect being by neflamapimod, as there would be no reason for treatment effects due to chance to segregate by the presence or absence of AD co-pathology.

All this being said, there are no established plasma, cerebrospinal or neuroimaging-based biomarkers of BFCN pathology and/or function. In addition, p38 MAPK has a wide number of functions in the CNS and is implicated in the pathogenicity of neurodegenerative processes outside of cholinergic degeneration[58]. Therefore, our results do not exclude neflamapimod having pharmacological activity on other neuronal or cell types (e.g., microglia) and biological effects other than those on endolysosomal dysfunction (e.g., on neuroinflammation), which may contribute to its clinical activity. Indeed, p38 MAPK is traditionally considered to be a target for reducing inflammation in the CNS, through reducing cytokine production from microglia and astrocytes[59]. However, in neurons, where p38α expression is low in the healthy state, its expression is increased under conditions of cellular stress and disease[27]; indeed in the Ts2 mouse model, phospho-p38α levels are higher than that in wild-type mice. And, more recent studies in animal disease models with the gene for the alpha isoform (MAPK14) knocked out specifically in neurons[28,48,49], or using specific p38α inhibitors[22], argue that neuronal p38α is also a relevant therapeutic target[58]. Further, as discussed below, the plasma drug concentrations achieved in the current clinical study are lower than that affecting cytokine production but consistent with that required for a neuronal effect[23,24].

While, at the start of the clinical study, 40 mg BID administered to lower-weight patients and 40 mg TID administered to higher-weight patients were anticipated to lead to similar results, exploratory analyses of the 40 mg TID recipients separately showed greater treatment effects relative to placebo, including improvement relative to placebo on the NTB and Attention Composites, compared to those seen in the analysis that included the 40 mg BID recipients. Although these results with twenty 40 mg TID participants require confirmation in a larger clinical study, the differential in treatment outcomes with 40 mg BID vs. 40 mg TID is consistent with the current understanding of the pharmacokinetics and pharmacokinetic-pharmacodynamic (PK-PD) relationship of neflamapimod. In terms of pharmacokinetics, prior assumptions regarding a weight effect were proven false in a

population pharmacokinetic analysis of the AD study[25] and the plasma drug levels with 40 mg BID were 30% lower than originally expected in the current study, despite the weight-dependent dosing regimen (i.e. TID in ≥80 kg and BID in <80 kg participants). Moreover, pharmacokinetic modeling indicates that the combined effect of a 50% increase in daily dose levels and the more frequent dosing frequency lead to a two-fold higher plasma trough drug concentration ($C_{trough}$) with 40 mg TID (median $C_{trough}$ = 6 ng/mL), compared with 40 mg BID (3 ng/mL). Of note, $C_{trough}$ was the pharmacokinetic parameter that best correlated to clinical efficacy in the AD trial, with evidence of a threshold effect for efficacy. Specifically, while 40 mg BID (only dose level evaluated in the AD trial) did not demonstrate clinical efficacy, there was evidence for clinical efficacy in the ~25% of neflamapimod participants in which 40 mg BID achieved a $C_{trough}$ > 4 ng/mL[25]. Finally, the average plasma drug concentration achieved with 40 mg TID was equivalent to the $EC_{50}$ for pharmacological activity in cellular systems (other than for anti-inflammatory effects, for which the $EC_{50}$ is two-fold higher)[23,24], ~10 ng/mL, and the estimated plasma drug exposure and $C_{trough}$ with 40 mg TID are similar to those achieved in preclinical pharmacology studies in the rat, the species in which there is sufficient pharmacokinetic information to make a comparison to human pharmacokinetics.

In the clinical study, no improvement was seen on the cognitive endpoint most dependent on normal hippocampal function, the ISLT, which measures episodic memory, including at the 40 mg TID dose level. This contrasts with the apparent effects of 40 mg TID on the Attention Composite and on the NTB composite, where the cognitive domains assessed are considered to be more closely associated with the cholinergic system. The findings are consistent with cholinesterase inhibitors in AD showing pro-cognitive effects on attention[60,61] while showing minimal to no discernible effects on recall outcome measures that are similar to the ISLT[61,62].

As the great majority of individuals with DS develop early onset AD (EOAD) and there is increased APP expression in both human DS and in the DS mouse, the Ts2 mouse model is regarded as a model for EOAD. While our clinical trial was conducted in patients with DLB, EOAD, and DLB both have major pathology in the basal forebrain cholinergic system and similar cortical atrophy patterns[55,57,63–65]. Furthermore, α-synuclein, the protein constituent of Lewy bodies, impairs retrograde axonal transport and BDNF signaling, in association with Rab5 and Rab7 accumulation[66], and reducing endogenous α-synuclein in an APP transgenic mouse decreases Rab5 protein levels and prevents degeneration of cholinergic neurons[67]. Conversely, in mouse models of DLB, amyloid beta (Aβ) plaques promote seeding and spreading of α-synuclein[68] and immunotherapy against Aβ and α-synuclein were additive against cholinergic fiber loss[69]. This literature, combined with our findings, suggests that, though the initiating factors may be different in EAOD and DLB, there is a common pathogenic process to cholinergic degeneration involving Rab5.

In conclusion, the preclinical and clinical results are consistent with neflamapimod being pharmacologically active against BFCN dysfunction, providing a foundation for hypothesis-testing (confirmatory) clinical evaluation of neflamapimod in diseases, such as DLB, with clinically consequential pathology in the basal forebrain. Beyond implications specific to neflamapimod, our findings inform on the impaired NGF-signaling based pathogenic model of cholinergic degeneration and the Ts2 DS mouse as a translational platform for target validation, for drug discovery and development, and for obtaining preclinical proof-of-concept support for therapeutic approaches to treating basal forebrain cholinergic dysfunction and degeneration associated with a range of neurologic disorders.

## Methods
### Preclinical study design
Mouse experimentation and animal care were approved by the Institutional Animal Care and Use Committee (IACUC) of the Nathan S.

Kline Institute. Ts2 (Stock No. 004850) mice, and wild type breeding partner (B6EiC3SnF1/J, Stock No. 001875) from the same colony were obtained from The Jackson Laboratory (Bar Harbor, ME). The protocol for drug dosage and oral gavage procedure was approved by the IACUC committee of Nathan S. Kline Institute (NKI) and renewed every three years, with the current protocol number AP2021-685-NKI.

The test compound neflamapimod (manufacturer lot number M10140) and instructions for formulation preparation, route of administration and dosing volume were provided by EIP Pharma, Inc. Both wild-type (2 N) and Ts2 mice were treated for four weeks, twice-daily by oral gavage at a 5 ml/kg dosing volume with vehicle (1% (w/v) Pluronic F108, Sigma) or neflamapimod [for determination of ChAT-immunoreactive BFCNs $n$ = 10 (2N-vehicle), $n$ = 11 (2N-NFMD), $n$ = 14 (Ts2-vehicle) and $n$ = 11 (Ts2-NFMD)]. Neflamapimod was freshly dissolved in 1% (w/v) Pluronic F108 at a final concentration of 0.6 mg/ml to administer it at 3 mg/kg. Treatment was initiated at 4.7–6.4 months of age for 2 N (control) and Ts2 mice, when endosomal pathology is evident and cholinergic neuronal loss is developing in Ts2 mice. Ahead of treatment initiation, treatment allocation was by the animal handlers by separating into different cages for the different treatment groups by matching the age and sex as best as possible to reduce the possible variation. Mice were housed in the NKI Animal facility and kept under 12 h day and night cycle, at temperatures at approximately 70° Fahrenheit (±2°) and humidity kept between 40% and 60%. To assess the health status of the animals throughout the studies and ensure their suitability for use in research, their general health status was monitored daily, and body weight was determined weekly.

One hour after either the terminal vehicle or neflamapimod treatment, mice were euthanized by displacement of air with 100% $CO_2$ and decapitated within 5 minutes. Whole mouse brains were collected and split into two brain hemispheres. The hippocampal sections were dissected out for electric physiology testing, while the remainders of the hemibrains were stored at −80 °C for further biochemistry analysis. For immunocytochemistry analysis, mice were anesthetized with a mixture of ketamine (100 mg/kg body weight) and xylazine (10 mg/kg body weight) and transcardially perfused with 4% paraformaldehyde in 0.1 M sodium cacodylate buffer, pH 7.4 (Electron Microscopy Sciences EMS), post-fixed in the same fixative overnight (O/N) at 4 °C and brain tissues were sectioned into 40 μm thick slices with a vibratome (Leica VT1000S).

### Immunocytochemistry and western blot analysis of preclinical samples
Immunohistochemistry was performed on 40 mm thick vibratome sections[18,70] using commercial antibodies against Rab5-GTP (NewEast, 26911; 1:50), ChAT (Millipore Sigma; AB144; 1:250), Rab5 (Abcam; 18211; 1:1000) and visualized with either biotinylated (Vector Laboratories; 1:500) or fluorescence-conjugated secondary antibodies (Fisher Sci, 1:500) as previously described[18,45,71]. Briefly, sections were washed three times with antibody dilution buffer containing 1% bovine serum albumin (BSA, Sigma), 0.05% saponin (Sigma), 1% normal horse serum (NHS, Thermo Fisher) in Tris-buffered saline (TBS), blocked with 20%NHS in TBS for one hour at room temperature, before incubated with primary antibodies overnight (O/N) at 4 °C. Confocal image were collected using Zeiss LSM880 laser confocal microscope and Zen 2.1-Sp3 software[72].

Morphometric analysis of Rab5-GTP puncta including intensity, number, average size and total area was determined by Fiji/ImageJ 2.3.0 (https://imagej.net/Fiji). Two male and two female mice were used and 20–30 neurons from each mouse were quantified and the means from each mouse were compiled.

For ChAT staining, sections were treated with 3% $H_2O_2$ before blocking, and incubated with anti-ChAT antibody (Millipore Sigma; AB144; 1:250). Diaminobenzidine (DAB) was visualized by incubating with biotinylated secondary antibody (1:500, Vector Laboratories) and Vectastain ABC kit (Vector Laboratories).

For protein analyses, mouse brain tissues were homogenized and western blot analyses were performed with antibodies against APP (c1/6.1; 1:1000), bCTF (M3.2, 1:250)[18], BACE1 (Rockland; 200-401-984; 1:500), MAPKAPK-2 (MK2; Cell Signaling; 12155; 1:500), phospho-MK2 (Cell Signaling; 3007; 1:500), p38 MAPK (p38α; Cell Signaling; 9218; 1:500), phosphor-p38 (Santa Cruz; 166182; 1:500), MNK1 (Cell Signaling; 2195; 1:500), pMNK1 (Cell Signaling; 2111; 1:500), β-actin (Santa Cruz Biotechnology; sc-47778; 1:2000). All the secondary antibodies for western blot analyses were used according to the manufacturer's recommendations (Jackson ImmunoResearch Laboratories, PA). A digital gel imager (Syn-gene G:Box XX9) was used to capture the ECL images and band intensities were quantified with Fiji/ImageJ 2.3.0 using β-actin as an internal control. Sizes of Rab5+ endosomes were determined by Image J, as previously described[18,45,71].

### Stereological counting of MS-ChAT+ neurons in Ts2 and wild-type mice

The numbers of ChAT-immunoreactive BFCNs in the MSN region were determined using the optical fractionator method[73] using ImageJ software controlling a Nikon Eclipse E600 microscope with a 100x oil immersion objective[18]. The MSN was sampled dorsal to the ventral edge of the anterior commissure, in every third consecutive vibratome section rostral to the decussation of the anterior commissure. ChAT+ cells were sampled with a grid of optical dissector counting sites. At each site a z-stack of six images with 1.5 micron spacing was saved to a computer. Counting boxes were drawn onto z-stacks with an upper guard zone of 1.5 microns (1 slice), a counting box of 4.5 microns, and a lower guard zone of the remaining slices. Cell counts were corrected for z-axis section thickness measured in triplicate on each section. Dissector size and sampling density were adjusted so that the coefficient of error was consistently <0.1 in all brains.

### Electrophysiology of Ts2 and wild-type mice treated with vehicle or neflamapimod

Six to 7-month-old 2 N and Ts2 mice, either treated with vehicle or with neflamapimod for 4 weeks ($n >> = 5$ for 2N-veh, 2N-NFMD and Ts2-NFMD, $n >> = 4$ for Ts2-veh) were used to measure long term potentiation (LTP) in CA1 hippocampal regions, as previously described[40,45]. The mice were sacrificed by cervical dislocation followed by decapitation and hippocampi were removed immediately, then transverse hippocampal slices (400 mm) were cut and placed in a recording chamber which was filled with artificial cerebrospinal fluid (ACSF, with 124 mM NaCl, 4.4 mM, KCl, 25 mM NaHCO$_3$, 2 mM MgSO$_4$, 10 mM glucose) at 29 °C and maintained consistently at 95% O$_2$ and 5% CO$_2$. Field EPSPs (fEPSPs) were recorded by placing both the stimulating and recording electrodes in hippocampal CA1 stratum radiatum, while basal synaptic transmission (BST) was determined by plotting the stimulus voltages over the slopes of fEPSPs. LTP was induced using theta-burst stimulation (4 pulses at 100 Hz, with the bursts repeated at 5 Hz, and each tetanus including three 10 burst trains separated by 15 s) and responses were recorded for 2 h and measured as fEPSP slope expressed as percentage of baseline.

### Preclinical evaluation of behavioral outcomes

Open field and novel object recognition (NOR) tests were performed according to previous publications[45,74]. A multiple unit open field maze with four activity chambers was used and activity of the mouse was recorded with a digital video camera linked to a computer. For NOR, 10 mins of exploration time was used for the analysis with CowLog 3, an opensource software[75]. Six to seven month-old 2 N and Ts2 mice ($n >> = 8$ for both) were tested before and after 4 weeks of neflamapimod treatment. The results were presented as Recognition Index (RI), which was defined as time exploring the novel object divided by the sum of time exploring both the novel and familiar objects as previously defined[45], with all the objects having similar textures and sizes but distinctive shapes. Between trials, the objects were cleaned with 10% ethanol.

For the open field test, 6 mins exploration time was used for the analysis with ToxTrac[76] and the following parameters, speed, travel distance and percentage of time spend in thigmotaxis (all edges of the chamber), were determined for each mouse.

### Statistical analysis of the preclinical study

All quantitative data were subjected to two-tailed unpaired Student's t-test for single comparison, and one-way ANOVA analysis for multiple comparisons, with post-hoc Tukey's analysis, using GraphPad Prism 8.0.1. Data are represented as bar graphs showing mean ± SEM with individual data point for each of mouse in the study group. For data involving fewer than 6 repeats, mean ± SEM were represented in bar graphs with individual data points. Statistical significance is represented by asterisks *$p \le 0.05$, **$p \le 0.01$, ***$p \le 0.001$. No statistical methods were used to predetermine sample sizes, but our sample sizes were chosen based on similar data in previous publications[16,18,23,40,45,74]. Based on the prior results, and preliminary data from a pilot study ($n >> = 3$), a sample size of 9 per treatment group was selected for the main study. Source data files are also included to show the raw data and sex distribution of the mice used for each figure.

### Clinical study design

Study EIP19-NFD-501 was a Phase 2, multicenter, randomized, double-blind, placebo controlled, proof-of-principle study of neflamapimod 40 mg or matching placebo conducted at 24 centers, 22 in the US and 2 in the Netherlands (see Acknowledgements for list of investigators). The first participant was enrolled on 30 September 2019 and the last visit occurred on 14 July 2020. The study was conducted in accordance with Good Clinical Practice guidelines and the Declaration of Helsinki. Applicable local/central ethics committee or IRB approvals were obtained. All participants provided written informed consent and were not compensated. IRB/Ethics approvals provided by Copernicus Group IRB (CGIRB, Cary, NC), Western Institutional Review Board (WIRB, Puyallup, WA), Mayo Clinic Institutional Review Board (Rochester MN), Columbia University Medical Center Institutional Review Board (New York, NY), Cleveland Clinic IRB (Cleveland, OH), and Foundation Beoordeling Ethiek Biomedisch Onderzoek (BEBO, Assen, the Netherlands). The trial was registered at clinicaltrials.gov as NCT04001517 on 28 June, 2019 and in the EU Clinical Trials Register on 26 June, 2019 with EudraCT Number of 2019-001566-15. The study protocol and statistical analysis plan (SAP) are available at https://clinicaltrials.gov/ct2/show/study/NCT04001517. The conduct of the study and preparation of the manuscript are in accordance with ICJME guidelines.

Participants included men and women aged ≥55 years with probable DLB, consistent with current consensus criteria[29], specifically at least one core clinical feature (fluctuating cognition, visual hallucinations, REM sleep disorder, and/or parkinsonism) and a positive DaTscan™ [30]; an MMSE score of 15–28, inclusive; and who were receiving cholinesterase inhibitor therapy (having received such therapy for greater than 3 months and on a stable dose for at least 6 weeks at the time of randomization) during Screening were eligible. Per the protocol, if the participant had a negative DaTscan™, but had a historical PSG-verified RBD, the participant qualified (six participants qualified on this basis). In addition, if a patient qualified on the basis of a positive DaTscan™ and only one core clinical feature, that core clinical feature could not be parkinsonism. Determination of whether the participant met eligibility criteria, including whether at least one core clinical feature was present, was made by the site principal investigator.

Following completion of informed consent procedures, participants entered the Screening phase of the study. One to two Screening visits were planned, during which safety screening measures were undertaken, practice cognitive tests were performed, and the required

diagnosis and cognitive impairment was confirmed. All screening assessments were to be conducted within 21 days of Day 1 (first dose of study drug), with the extension to 35 days allowed if a DaTscan™ was required to determine study eligibility.

After eligibility was confirmed and before the first dose of study drug, participants were randomized via an interactive web-based response system provided by Suvoda LLC, Conshohocken PA, USA, which automatically generated the random code. The neflamapimod: placebo randomization ratio was 1:1 and stratified by International Shopping List Test Total Recall score at Baseline (<21 vs. ≥21), i.e., by whether patients have an episodic memory defect at baseline or not. The randomization block size was 6 within each strata.

Neflamapimod and placebo capsules were identical in appearance. Capsules were in blister strips and packaged into kits containing two weeks supply. Each kit was labeled with a unique identifier assigned by the vendor and the specific kit(s) to be administered to a participant at a given visit was allocated by the interactive web-based response system. Patients, caregivers, investigators, outcome assessors and all other personnel involved in the conduct of the clinical trial were blinded to treatment allocation.

Participants s followed the BID regimen if weighing <80 kg or the TID regimen if weighing ≥80 kg. Doses were taken within 30 min following a meal or snack (i.e., morning, mid-day (TID only), and evening meals) at least 3 h apart and at approximately the same times each day throughout the study. Dosing started on Day 1, following completion of all Baseline procedures. During the treatment period, subjects attended study center visits on Days 14, 28, 56, 84, and 112. Due to the COVID-19 global pandemic, some of these visits were permitted to be held remotely due to local restrictions to travel and in-clinic visits.

As this was an exploratory trial and there was no prior experience with neflamapimod in patients with DLB upon which to base assumptions of treatment effect, no formal sample size calculation was performed. However, based on prior experience with NTB in clinical studies, 40 subjects per treatment arm was thought to provide a reasonably robust first assessment of whether neflamapimod improves cognitive function in patients with DLB. Accordingly, the study protocol specified that a total of approximately 80 subjects were planned to be enrolled, of whom 40 were planned to receive neflamapimod and 40 were planned to receive placebo.

### Plasma ptau181 measurement

Plasma p-Tau181 was measured in the clinical chemistry laboratory at the VU Medical Center according to the kit instructions (p-Tau181 V2 kit, Quanterix, Billerica, USA). In-house quality control plasma samples were run in duplicates at the start and the end of each plate to determine the within-run and between-run variations.

### Clinical study outcome measures

The primary objective was to evaluate the effect of neflamapimod on cognitive function as assessed in a study-specific Neuropsychological Test Battery (NTB) comprised of four computerized tests from the Cogstate® cognitive testing battery (Detection, Identification, One Card Learning, One Back) and two verbal fluency tests (Letter Fluency Test, Category Fluency Test) that were recorded on paper.

Secondary objectives included to (1) evaluate effects on cognition and function, as assessed by the Clinical Dementia Rating Scale sum-of-boxes (CDR-SB); (2) evaluate effects on motor function, as assessed by the Timed Up and Go test; (3) evaluate effects on episodic memory, as assessed by the International Shopping List Test (ISLT); (4) evaluate the effects on general cognition, as assessed by the Mini Mental Status Examination (MMSE); and (5) evaluate effects on select domains of the 10-item Neuropsychiatric Inventory (NPI-10), including depression (dysphoria), anxiety, hallucinations, and agitation/aggression. At study start, quantitative EEG was included as an additional endpoint but results are not reported due to the impact of COVID-19-related

lockdowns on collection of this measure. With the closure of EEG laboratories to elective evaluations, only approximately one-third of patients were able to have both baseline and week 16 EEG recordings.

### Clinical study statistical analysis

Except for the cognitive tests within the Cogstate Battery (four within NTB, and the ISLT) all data were captured by Worldwide Clinical Trials via an electronic data capture (EDC) and, after all queries were resolved and database was locked, SAS datasets were provided to Anoixis corporation for statistical analysis. The data from the Cogstate tests were processed at Cogstate and provided directly to the study statistician at Anoixis Corporation (Natick, MA, USA). S-PLUS (Version 8.2), R (Version 3.6.3 or higher) or SAS (Version 9.3 or higher) were utilized for the statistical analysis. The Efficacy Population used for efficacy analyses, consistent with the intent-to-treat (ITT) principles included all participants who have a baseline and at least one on-study assessment of the efficacy parameter (i.e., modified intent-to-treat population, mITT).

Due to the various scales among different tests in the NTB and the need to have equal weights for deriving composite score metrics performance on each, the tests were standardized relative to baseline data from all randomized subjects (i.e., the individual test core was converted to a z-score by subtracting the study sample mean at baseline from the score and dividing by the standard deviation (SD) of the study sample baseline). The NTB composite z-score was calculated as the average of the z-scores from the component of the six individual tests in the composite. For the purposes of facilitating further understanding of the drug effects on the cholinergic system, an exploratory Attention Composite endpoint, made up of the two tests in NTB that evaluate information processing speed, the Identification and Detection tests, was calculated and evaluated.

As pre-specified in the protocol, the analyses of all efficacy endpoints (primary, secondary, and exploratory) used a Mixed Model for Repeated Measures (MMRM) analysis method, without imputation of missing values, with change from baseline as the dependent variable, and with a fixed effect on treatment extended to baseline composite score and study visit as covariates. There was a fixed effect on treatment that was extended to baseline composite score and study visit. The interaction term (i.e., scheduled visit by treatment) was considered. Per the protocol, as there was no stated hypothesis for treatment effect, no p-values are provided and instead differences from placebo from the MMRM analysis, with 95% confidence intervals, are reported. In addition, Cohen's d effect size for each outcome measure was calculated and is reported.

### Reporting summary

Further information on research design is available in the Nature Research Reporting Summary linked to this article.

## Data availability

The preclinical data associated with Figs. 1–3 are provided in the Source Data file. Deidentified individual participant baseline data contained in Table 1 and on-study data from placebo recipients for the endpoints contained in Table 2 in this article will be made available upon request to the corresponding author (JJA) to investigators whose proposed use has been approved by an independent review committee, beginning 9 months, and ending 36 months after publication. The study protocol and statistical analysis plan (SAP) are available at https://clinicaltrials.gov/ct2/show/study/NCT04001517. Source data are provided with this paper.

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

## Acknowledgements

AscenD-LB Clinical Study investigators: USA: K. Amadeo (U. of Rochester, Rochester, NY), G. Baras (Elite Clinical Research, Miami, FL), K. Bell (Columbia U. Medical Center, New York, NY), C. Bernick (U. of Washington, Seattle, WA), B. Boeve (Mayo Clinic, Rochester, NY), N. Bohnen (Michigan Medicine, Ann Arbor, MI), S. Gomperts (Massachusetts General Hospital, Charlestown, MA), S. Holden (U. of Colorado, Aurora, CO), D. Kaufer (U. of North Carolina, Chapel Hill, NY), S. Kesari (Pacific Neuroscience Institute, Santa Monica, CA), A. Khan (Northwest Clinical Research Center, Bellevue, WA), I. Litvan (UC San Diego Health, San Diego, CA), S. Losk (Summit Research Network, Portland, OR), R. Pahwa (U. of Kansas Medical Center, Kansas City, KS), S. Pugh (Inland Northwest Research, Spokane, WA), J. Quinn (Oregon Health and

Science University, Portland OR), A. Ritter (Cleveland Clinic, Las Vegas, NV), B. Shah (UVA Health, Charlottesville, VA), D. Scharre (Ohio State Neurological Institute, Columbus, OH), M. Serruya (Jefferson U. Hospitals, Philadelphia, PA), B. Tousi (Cleveland Clinic, Cleveland, OH); Netherlands: P. Dautzenberg (Brain Research Center (BRC) – Den Bosch, Den Bosch), A.W. Lemstra (Brain Research Center (BRC) – Amsterdam, Amsterdam). The preclinical study was supported by NIH grants P01AG017617 and R01AG062376 grants to RAN. BSB and SS are supported by NIH grant R01 AA019443. The clinical study was funded by EIP Pharma, Inc (Boston, MA). EIP Pharma, with input from the lead clinical investigators (SNG, NP, AWL, PS, JEH), designed the clinical study and wrote the clinical study protocol. EIP Pharma, primarily through JJA, evaluated the results of the statistical analysis and played a major role in writing the manuscript. The authors are very grateful to all patients and caregivers who participated in this study and to the many staff members at the clinical sites for their dedication and commitment to the clinical study. We also acknowledge the study project teams at EIP Pharma and Worldwide Clinical Trials, as well Dr. Susan Doctrow for editorial and technical assistance in preparing the manuscript and Dr. Sylvie Grégoire for critical reading of the manuscript.

## Author contributions

The Ts2 mouse study was conceived of by R.A.N. and designed by Y.J. with input from R.A.N., J.J.A. and U.A.G. Y.J. supervised by R.A.N., organized and led the conduct of the preclinical study, and conducted the majority of the collection and analysis of the data from that study. P.H.S., S.D., S.M., D.-S.Y., M.P., and J.P. conducted clinical aspects (animal husbandry, drug administration, etc.) of the Ts2 mouse study. PHS also conducted the behavioral studies, J.P. performed the western blot analysis of brain tissue, and CB and JFS conducted the immunocytochemistry and ChAT+ neuronal analysis. S.S. and B.S.B. conducted and analyzed the LTP studies. A.P. collected the Rab5-GTP images and provided scientific input to various aspects of the Ts2 mouse study. The clinical study was conceived of by J.J.A. and designed by J.J.A., P.S., N.D.P., and J.E.H.; with input from P.M. and K.B. The NTB was designed by J.E.H. and P.M. constructed the statistical approach to analyzing the NTB composite (i.e. construct of z-scores). J.J.A. was the sponsor lead for the clinical study, and the study was conducted under the clinical project leadership of A.G. and K.B. S.N.G. and A.W.L. were clinical investigators in the study and lead investigators in the US and Netherlands, respectively. The plasma ptau181 evaluations were conducted in the laboratory of CET. H.-M.C. conducted all the statistical analyses of the clinical study data. The manuscript was largely drafted by Y.J. and J.J.A., under the supervision and leadership of R.A.N. and it was finalized by J.J.A. with substantial input from R.A.N., U.A.G., S.N.G., and P.M. The clinical sections of the manuscript were also reviewed by N.P., A.W.F., and P.S.

## Competing interests

J.J.A, A.G., and K.B. are employees of EIP Pharma, the sponsor of the clinical study. J.J.A. is also founder of and has stock ownership in EIP Pharma. U.A.G. receives compensation as a scientific consultant to EIP Pharma. S.N.G. has served on Advisory Boards of Jannsen, Acadia, and Sanofi, has received consulting fees from EIP Pharma, and has received funding from the NIH, the DOD CDMRP, the Michael J. Fox Foundation, the FFFPRI, and the Lewy Body Dementia Association. N.D.P. is CEO and co-owner of Brain Research Center. P.M. is a full-time employee at Cogstate Ltd. J.E.H. reports receipt of personal fees in the past 2 years from Actinogen, AlzeCure, Aptinyx, Astra Zeneca, Athira Therapeutics, Axon Neuroscience, Axovant, Bial Biotech, Biogen Idec, BlackThornRx, Boehringer Ingelheim, Brands2life, Cerecin, Cognito, Cognition Therapeutics, Compass Pathways, Corlieve, Curasen, EIP Pharma, Eisai, G4X Discovery, GfHEU, Heptares, Ki Elements, Lundbeck, Lysosome Therapeutics, MyCognition, Neurocentria, Neurocog, Neurodyn Inc, Neurotrack, the NHS, Novartis, Novo Nordisk, Nutricia, Probiodrug, Prothena, Recognify, Regeneron, reMYND, Rodin Therapeutics, Samumed, Sanofi, Signant, Syndesi Therapeutics, Takeda, Vivoryon Therapeutics and Winterlight Labs. In addition, he holds stock options in Neurotrack Inc. and is a joint holder of patents with My Cognition Ltd. C.E.T. has a collaboration contracts with ADx Neurosciences, Quanterix and Eli Lilly; performed contract research or received grants from AC-Immune, Axon Neurosciences, Bioconnect, Bioorchestra, Brainstorm Therapeutics, Celgene, EIP Pharma, Eisai, Grifols, Novo Nordisk, PeopleBio, Roche, Toyama, and Vivoryon; and has had speaker contracts for Roche, Grifols, and Novo Nordisk. P.S. has received consultancy fees (paid to the institution) from AC Immune, Brainstorm Cell, EIP Pharma, ImmunoBrain Checkpoint, Genentech, Novartis, Novo Nordisk. P.S. is also principal investigator of studies with AC Immune, FUJI-film/Toyama, UCB, and Vivoryon; and is an employee of EQT Life Sciences (formerly LSP). The remaining authors declare no competing interests.

## Additional information

[1]Center for Dementia Research, Nathan S. Kline Institute for Psychiatric Research, 140 Old Orangeburg Rd, Orangeburg, NY 10962, USA. [2]Department of Psychiatry, NYU Grossman School of Medicine, 550 First Ave, New York, NY 10016, USA. [3]EIP Pharma Inc, 20 Park Plaza, Suite 424, Boston, MA 02116, USA. [4]Massachusetts Alzheimer's Disease Research Center, 114 16th Street, Massachusetts General Hospital, Charlestown, MA 02129, USA. [5]Cogstate Ltd, Runway East Borough Market, 20 St. Thomas St, London SE1 9RS, UK. [6]Alzheimer Center Amsterdam, Amsterdam Neuroscience - Neurodegeneration, Amsterdam UMC at VUmc, De Boelelaan 1118, 1081 HZ Amsterdam, the Netherlands. [7]Brain Research Center, Cronenburg 2, 1081 GN Amsterdam, the Netherlands. [8]Molecular Imaging and Neuropathology Area, New York State Psychiatric Institute, 1051 Riverside Dr., New York, NY 10032, USA. [9]Department of Psychiatry, Columbia University, 1051 Riverside Dr., New York, NY 10032, USA. [10]Anoixis Corporation, 214 N. Main St. #104, Natick, MA 01760, USA. [11]Metis Cognition Ltd, Park House, Kilmington Common, Warminster BA12 6QY, United Kingdom. [12]Department of Cell Biology, NYU Grossman School of Medicine, 550 First Ave, New York, NY 10016, USA. [13]These authors contributed equally: Ying Jiang, John J. Alam. ✉e-mail: jalam@eippharma.com; ralph.nixon@nki.rfmh.org

