## [Peer Review File · Nature Communications]

Preclinical and Randomized Clinical Evaluation of the p38 α Kinase Inhibitor Neflamapimod for Basal Forebrain Cholinergic DegenerationThis manuscript has been previously reviewed at another journal that is not operating a transparent peer review scheme. This document only contains reviewer comments and rebuttal letters for versions considered at *Nature Communications*.

REVIEWERS' COMMENTS

Reviewer #3 (Remarks to the Author):

Overall, this reviewer was pleased with the revision. There were a few minor points that need attention:

P9 and Table 2. This notation was confusing – ($p > 0.2$, difference to placebo = 0.04, 95% CI: -0.11, 0.19). It might be helpful to change “difference to placebo” to “drug-placebo difference”.

P 10, 4 lines from bottom. Some words are missing between “intimately” and “atrophy”, probably “related to?”

This reviewer found the speculations about the effects of Neflamapimod on gait on p 15-16 to be excessive, particularly in relation to the next paragraph that went off in another direction regarding the subset of participants with elevated ptau181. On the former grounds the authors invoke motoric functionality for basal forebrain and on the latter AD pathology. Both discussions could be shortened and less speculative.

Reviewer #4 (Remarks to the Author):

This is an interesting article containing reports of a experimental and clinical evaluation of Neflamapimod for dementia like syndromes that has undergone extensive previous review. Authors have responded appropriately to previous comments. While the article is long it is the sum of two reports and it is generally clear where readers can find specific items of information. I have therefore kept my comments as limited as possible.

1. Suggest put Neflamapimod somewhere in the title as I think this helps readers and also helps to search for articles on this agent
2. Table 2 Efficacy outcomes: I would prefer to see both the baseline values for placebo and active groups (provided) as well as the follow up measurements in each group (not provided). Mean differences can be provided as well.
3. Suggest place Figure 3a and b from supplemental appendix in the main manuscript as readers will want to see this and include a clear explanation of the findings in a figure legend
4. Please ensure that supplemental figures are properly explained e.g. Figure 1 does not really provide information for reader to interpret the data e.g. is does a negative change mean an improvement in hallucination severity? Please check all supplemental figures carefully to determine whether more information is required for the reader. The main figures appear to be well explained.

Reviewer #5 (Remarks to the Author):

The authors have revised the manuscript to reflect the treatment approach based on therapeutic mechanisms related to basal forebrain Cholinergic degeneration as tested in rodent models and then translated to a clinical trial in DLB, a disorder in which degeneration of these cholinergic neurons is a prominent pathological feature.

In response to reviewer #2 in particular, the authors have responded clearly to points raised, as follows:

1. Title: it would be clearer to have DLB in the title, but it is clearly mentioned in the Abstract, which is helpful. In the Abstract, it states that BFCN degeneration is the primary driver of disease expression in DLB. There is more widespread pathology in DLB, including involvement of other brainstem areas and temporal lobe and neocortical areas by a-synuclein pathology. Using a consistent term that acknowledges this, e.g., an “important driver” would be clearer and should be applied throughout the manuscript. The presence of additional regional pathology in DLB (and in AD) may make it harder to demonstrate a therapeutic benefit in clinical trials in DLB and AD, compared to doing so in a mouse model.
2. Introduction and background: once again, using the term ‘important driver’ would be more appropriate
3. Authors have clarified use of a vehicle in the mouse model as a control.
4. On page 4, it would be good to mention that the AD trial did not show a benefit, although

relatively low doses were tested.

5. Criteria for patient diagnosis and selection for the trial are now described

6. Use of DaT as a diagnostic criterion: this is partly discussed in the response. Although it helped to select a diagnostically more homogeneous group of people with DLB, it may also have selected for a subgroup where cholinergic interactions with dopamine deficits could affect gait and balance and favor the positive findings regarding the TUG test.

7-11. All points are adequately discussed in the response.

12. Adequately described in the table listing SAEs.

There is a typo on p17 – 6th line from bottom reads “40 mg tid vs 40 mg tid”

Reviewer #6 (Remarks to the Author):

This is the revised version of the manuscript by Jiang et al that investigated the potential use of the anti-inflammatory drug neflamapimod in a mouse model of Down syndrome and as a phase II clinical trial in individuals with DLB. The authors report that neflamapimod which is a p38 alpha inhibitor originally developed for chronic arthritis corrected alterations in Rab5, endosomal pathology and cholinergic neurons in a mouse trisomy model. In DLB, a 16-week-treatment clinical trial the drug failed to show an effect in the clinical primary outcome measures, however some modest changes were observed in some secondary outcome measures. The authors conclude that the results of the preclinical mouse study and patients with DLB with this compound advance our understanding of the role of p38, endosomes and cholinergic degeneration.

The revised version of the manuscript has been extensively modified to account for several points raised by the reviewers relevant to the design of the clinical trial, outcomes measures, statistical analysis, power of the study, moreover the reviewers had several questions about the choice of the animal model and limited analysis performed. I was not involved in the original review of the manuscript but overall although the authors provide a long and detailed rebuttal that is helpful at explaining their interpretation of the data and how the preclinical and clinical studies interplay, however it is difficult to conclude due to lack of more detail how the two studies are fully connected mechanistically.

That is many of the reviewers' comments which request additional, and more detail data might stem from the design of the manuscript that in my opinion represents the skilled and interesting effort to combine two papers in one.

As the authors are well aware DLB is a complex synucleinopathy, certainly degeneration of cholinergic neurons is a component, but the DS model is not a model of DLB, there are better models driven by alpha synuclein pathology with cholinergic degeneration that mimic more reasonably certain aspects of DLB. Moreover, trying to connect Rab5, endosomal lysosomal function and p38 alpha in a model of DS and then translate that to DLB is not fully justified. It would have been more logical the connection if the trial was in individuals with DS, or if the preclinical studies would have been done in a KI, transgenic or pff model of DLB.

In addition, there is no data showing the mechanistic connection between Rab5 and p38 alpha that is primarily a glial cell kinase, while most of the endosomal pathology with Rab5 in human brains and AD models is neuronal. The authors have not shown convincingly that there is target engagement of neflamapimod in the CNS, this is a compound over 500 daltons with poor Lipinsky rule profile that have low BBB penetration. The other problem with neflamapimod is that it also inhibits other p38 isoforms and that this compound has failed in AD clinical trials, for these reasons more selective compounds with better CNS profile has been developed by some labs including the Van Eldik group.

POINT-BY-POINT RESPONSES TO REVIEWERS' COMMENTS

Reviewer #3

Overall, this reviewer was pleased with the revision. There were a few minor points that need attention:

P9 and Table 2. This notation was confusing – ($p > 0.2$, difference to placebo = 0.04, 95% CI: -0.11, 0.19). It might be helpful to change “difference to placebo” to “drug-placebo difference”.

Response: We have changed “difference to placebo” and “difference” to “drug-placebo difference” throughout the text and in Table 2. Thank you for this suggestion, as this will be a very helpful change for the reader.

P 10, 4 lines from bottom. Some words are missing between “intimately” and “atrophy”, probably “related to?”

Response: We have inserted “related to” in that sentence.

This reviewer found the speculations about the effects of Neflamapinod on gait on p 15-16 to be excessive, particularly in relation to the next paragraph that went off in another direction regarding the subset of participants with elevated ptau181. On the former grounds the authors invoke motoric functionality for basal forebrain and on the latter AD pathology. Both discussions could be shortened and less speculative.

Response: Based on multiple human clinical studies in PD and DLB published in the past 12 to 18 months which have correlated basal forebrain atrophy to gait dysfunction (references #8-10 in revised manuscript), as well functional connectivity studies (references #50, #53), that the basal forebrain has a direct role in motor control and function should no longer be considered “speculative”. Importantly, for the quality of gait in PD, the cholinergic deficit is more strongly predictive than the dopaminergic system, explaining to a great extent the lack of impact of dopaminergic agents on this aspect of the gait dysfunction in PD (reference #8, 9). With this understanding of the connection between the basal forebrain and motor function (specifically, gait), the Timed up and Go (TUG) results is a major part of the clinical result in our report with respect to a pharmacological effect of neflamapimod on the basal forebrain cholinergic system. We understand that gait dysfunction is only one component of the clinical presentation of DLB, but as the focus of our report is on the translation of the preclinical effect on the basal forebrain cholinergic system into the clinic, we believe that the length and depth of discussion of the TUG result is appropriate to the importance in relation to the conclusions of the report.

We do hypothesize, and clearly identify it as a hypothesis, that a distinct effect of neflamapimod on motor function in patients otherwise receiving cholinesterase inhibitors is the reason there is a more robust effect on the TUG and CDR-SB compared that on the cognitive endpoints, including the primary endpoint. However, as for the reader, the differential effect on these endpoints is a potential confounder to the main conclusion, putting forward a hypothesis for the discordance is a fair, and perhaps expected aspect of a Discussion section.

With respect to the discussion of the analysis of the results stratified by baseline plasma ptau181, it is indeed a distinct and separate discussion point (and thus, a separate paragraph). In the revised manuscript, we have added a newly published reference [reference #57, Kantarci, K., *et al.* Longitudinal atrophy in prodromal dementia with Lewy bodies points to cholinergic degeneration. *Brain Commun* **4**, fcac013 (2022)], that directly makes the case that the predominant pathology in DLB patients who do not have AD co-pathology is basal forebrain cholinergic degeneration. We agree with the Reviewer that the literature up to the point of our prior submission was more indirect on this point, but with this publication, the arguments within the ptau181 discussion become less speculative. We would also note that the presentation here is not in the main results section, for the very reasons cited by the reviewer, and are planning on a more detailed presentation of these results in a separate report. In the current report, there is a concise description of the top line result only.

Reviewer #4:

This is an interesting article containing reports of a experimental and clinical evaluation of Neflamapimod for dementia like syndromes that has undergone extensive previous review. Authors have responded appropriately to previous comments. While the article is long it is the sum of two reports and it is generally clear where readers can find specific items of information. I have therefore kept my comments as limited as possible.

1. Suggest put Neflamapimod somewhere in the title as I think this helps readers and also helps to search for articles on this agent

Response: We have inserted the word neflamapimod in the title and to stay to the word count limit, have removed “p38 MAPK inhibition”

2. Table 2 Efficacy outcomes: I would prefer to see both the baseline values for placebo and active groups (provided) as well as the follow up measurements in each group (not provided). Mean differences can be provided as well.

Response: The mean difference between placebo and neflamapimod and the 95% confidence interval of that difference from the statistical (MMRM) analysis is reported in Table 2 as the “Difference-On-Study”. Given this comment, the comment from Reviewer #3, and in accordance with the recommendation from Reviewer #3, we changed the descriptor to “Drug-placebo difference”, both in Table 2 and throughout the text.

In accordance with the protocol and the statistical analysis plan, the primary and secondary clinical endpoints were evaluated as reported in Table 2, i.e., as the mean difference between placebo and neflamapimod in the *change from baseline* (and not absolute values) using the mixed model for reported measures (MMRM) approach. Of note, the MMRM approach utilizes all timepoints on study and not just the end of study timepoint compared to baseline. In addition, the model that was utilized included baseline as a covariate, and so reported difference is adjusted for baseline values. As discussed in our first response, as a result of all these factors (change from baseline analysis, utilization of multiple timepoints, adjustment for baseline) the “follow up measurement” (i.e., the mean absolute value at the end of the study)

by group does not align well with the MMRM and in a study of this size and duration is not interpretable (i.e., is not helpful to the reader).

We agree with the reviewer that in larger studies of a longer duration, i.e., in phase 3, there is substantial clinical insight provided by the absolute values. However, in a phase 2a study of this size and duration, the change from baseline analysis is the standard and appropriate analysis, as well as the analysis pre-specified in the protocol. The major issue with the absolute values requested by the reviewer is that the inter-subject variability (i.e., standard deviation) at baseline is high relative to the change that occurs over the duration of the study. As a result, the variability in the absolute values masks any evaluation of differences that occur over time in shorter duration studies.

From a presentation standpoint, it is a little bit less of an issue for the primary endpoint because z-scores were utilized, which inherently normalizes to the baseline mean. As a result, the mean values at any time point are essentially the same as change from baseline. Nevertheless, for the main manuscript, the most appropriate presentation is what is provided in Table 2. We have provided the descriptive data in the Supplemental figure 3, including the absolute mean (+/- SEM) z-score at each time point on study (see also response to next comment) .

3. Suggest place Figure 3a and b from supplemental appendix in the main manuscript as readers will want to see this and include a clear explanation of the findings in a figure legend

Response: Supplemental figure 3 was in the original version of the manuscript but was moved into the Supplemental information at the request of, and strong recommendation of the editor that reviewed that original version. In retrospect, as discussed in the previous response, keeping the focus on the MMRM analysis in the main manuscript is the most appropriate approach. Also, the supplemental information is available to the reader who wants to go deeper. Importantly, the size and duration of the clinical study is such that it inherently was not designed to inform on clinical meaningfulness, which differences in group means could inform upon. Instead in a phase 2a study, the MMRM analysis is the most robust approach and Cohen's d effect size provides the best assessment of magnitude of the treatment effect relative to placebo.

4. Please ensure that supplemental figures are properly explained e.g. Figure 1 does not really provide information for reader to interpret the data e.g. is does a negative change mean an improvement in hallucination severity? Please check all supplemental figures carefully to determine whether more information is required for the reader. The main figures appear to be well explained.

Response: We have added explanatory notes or legends throughout the supplemental information. Regarding Supplemental Figure 1, there was an error that has now been corrected. The Reviewer is correct that the negative direction means improvement, while the arrow in the prior version indicated the opposite.

Reviewer #5:

The authors have revised the manuscript to reflect the treatment approach based on therapeutic mechanisms related to basal forebrain Cholinergic degeneration as tested in rodent models and then translated to a clinical trial in DLB, a disorder in which degeneration of these cholinergic neurons is a prominent pathological feature.

In response to reviewer #2 in particular, the authors have responded clearly to points raised, as follows:

1. Title: it would be clearer to have DLB in the title, but it is clearly mentioned in the Abstract, which is helpful. In the Abstract, it states that BFCN degeneration is the primary driver of disease expression in DLB. There is more widespread pathology in DLB, including involvement of other brainstem areas and temporal lobe and neocortical areas by a-synuclein pathology. Using a consistent term that acknowledges this, e.g., an “important driver” would be clearer and should be applied throughout the manuscript. The presence of additional regional pathology in DLB (and in AD) may make it harder to demonstrate a therapeutic benefit in clinical trials in DLB and AD, compared to doing so in a mouse model.

2. Introduction and background: once again, using the term ‘important driver’ would be more appropriate

Response: As discussed in more detail in the prior response, the major focus of the report is on the potential of neflamapimod to treat basal forebrain cholinergic (BFC) degeneration. DLB is very good context to evaluate effects on BFC degeneration, as is the Ts2 mouse model, but the link between them is the BFC degeneration. It is the reason why in this translational report that reports on both the mouse model and clinical study in DLB, we refer to the common aspect (BFC degeneration) in the title, and not to either DLB or the mouse model.

As recommended, we have changed “primary driver of disease expression” to “an important driver of disease expression” in the abstract and manuscript. To support that importance, we have incorporated a very recent publication that further adds granularity on the critical role of cholinergic degeneration in DLB, particularly in the early stages of the disease [reference #57, Kantarci, K., *et al.* Longitudinal atrophy in prodromal dementia with Lewy bodies points to cholinergic degeneration. *Brain Commun* 4, fcac013 (2022)].

3. Authors have clarified use of a vehicle in the mouse model as a control.

Response: Thank you for the comment.

4. On page 4, it would be good to mention that the AD trial did not show a benefit, although relatively low doses were tested.

Response: We discuss the AD trial in the Discussion section and explicitly state that the AD trial “did not demonstrate clinical efficacy”. As the AD trial did not report out prior the DLB clinical study and as such did not inform on the design of the latter (i.e., is not part of the background

for conducting the latter), we have not repeated that discussion in the introduction. Rather we have referred the reader to the discussion when we refer to the AD trial in the introduction.

5. Criteria for patient diagnosis and selection for the trial are now described

Thank you for the comment.

6. Use of DaT as a diagnostic criterion: this is partly discussed in the response. Although it helped to select a diagnostically more homogeneous group of people with DLB, it may also have selected for a subgroup where cholinergic interactions with dopamine deficits could affect gait and balance and favor the positive findings regarding the TUG test.

Response: We agree that DaT scan may have selected for patients with gait dysfunction. If indeed that were true, it would impact the clinical generalizability of the findings, but does not impact the interpretability with regard to the pharmacological activity of neflamapimod. And otherwise, the DaT scan would have achieved its objective of enriching for patient population that were aligned with the mechanism of action. With respect to clinical generalizability, the sensitivity of the DaT scan against autopsy confirmed DLB is ~80% (reference #30). That is the DaT scan excludes only 20% of the overall population (those with “minimal brainstem involvement”), while increasing the specificity of the diagnosis to greater than 90%. That is, the patient population included in the study was representative of the great majority of patients with DLB.

7-11. All points are adequately discussed in the response.

12. Adequately described in the table listing SAEs.

There is a typo on p17 – 6th line from bottom reads “40 mg tid vs 40 mg tid”

Response: Thank you for the above comments and the correction.

Reviewer #6 (Remarks to the Author):

This is the revised version of the manuscript by Jiang et al that investigated the potential use of the anti-inflammatory drug neflamapimod in a mouse model of Down syndrome and as a phase II clinical trial in individuals with DLB. The authors report that neflamapimod which is a p38 alpha inhibitor originally developed for chronic arthritis corrected alterations in Rab5, endosomal pathology and cholinergic neurons in a mouse trisomy model. In DLB, a 16-week-treatment clinical trial the drug failed to show an effect in the clinical primary outcome measures, however some modest changes were observed in some secondary outcome measures. The authors conclude that the results of the preclinical mouse study and patients with DLB with this compound advance our understanding of the role of p38, endosomes and cholinergic degeneration.

The revised version of the manuscript has been extensively modified to account for several points raised by the reviewers relevant to the design of the clinical trial, outcomes measures, statistical analysis, power of the study, moreover the reviewers had several questions about the choice of the animal model and limited analysis performed. I was not involved in the original review of the manuscript but overall although the authors provide a long and

detailed rebuttal that is helpful at explaining their interpretation of the data and how the preclinical and clinical studies interplay, however it is difficult to conclude due to lack of more detail how the two studies are fully connected mechanistically.

That is many of the reviewers' comments which request additional, and more detail data might stem from the design of the manuscript that in my opinion represents the skilled and interesting effort to combine two papers in one.

Response: The mechanistic connection between the preclinical and clinical study is that they evaluated neflamapimod in preclinical and clinical contexts, respectively, where there is robust basal forebrain cholinergic degeneration. Indeed, the Down Syndrome mouse models are the preclinical models where the basal forebrain cholinergic degeneration has been most extensively studied and the mechanisms underlying it are best understood. Moreover, as acknowledged by reviewers #2 and #3 in the original review, basal forebrain cholinergic degeneration and resulting cholinergic deficit are understood to play a major role in disease expression and progression. Further, as discussed in the next response, the underlying pathogenic mechanisms in each case are likely to be overlapping, specifically in both contexts being Rab5 dependent.

Additionally, we would re-iterate the statements at the beginning of our prior response related to the objective of the manuscript being an investigation of a pharmacological effect towards basal forebrain cholinergic degeneration:

“The intent in submitting to a translational journal, rather than a traditional clinical journal is not only to report encouraging findings on the treatment of DLB, a devastating disease with no cure or current treatment options, but to link the outcomes to strong preclinical support demonstrating a proof-of-mechanism for neflamapimod, the p38 α kinase inhibitory drug (NFMD) used in the study, as a novel therapy against basal forebrain cholinergic degeneration. While our clinical trial of a drug therapy for dementia with Lewy bodies (DLB) is a critical feature of the report, the translational implications of our bench-to-bedside study go well beyond the possibility of a DLB therapy by revealing a strategy relevant to treatment in a range of neurological diseases in which cholinergic neurodegeneration is clinically important.

It is correct that the original objective of the clinical trial was as an exploratory screening study to evaluate neflamapimod treatment effects on various clinical parameters in patients with DLB. However, the mechanistic preclinical studies were largely conducted during the clinical study, the results of which led to further support that neflamapimod therapeutically targets the basal forebrain cholinergic system. Importantly, during and after the conduct of the clinical trial, a range of translational study reports have transformed the understanding of basal forebrain cholinergic system function, as well the clinical correlates of basal forebrain pathology and/or dysfunction (reference 8-10, 50-53 in the manuscript). These advances allowed us to understand the clinical results in a more comprehensive light, leading to greater mechanistic insights into the pharmacological and biological actions of neflamapimod on basal forebrain

cholinergic dysfunction and degeneration in the context of DLB.”

As the authors are well aware DLB is a complex synucleinopathy, certainly degeneration of cholinergic neurons is a component, but the DS model is not a model of DLB, there are better models driven by alpha synuclein pathology with cholinergic degeneration that mimic more reasonably certain aspects of DLB. Moreover, trying to connect Rab5, endosomal lysosomal function and p38 alpha in a model of DS and then translate that to DLB is not fully justified. It would have been more logical the connection if the trial was in individuals with DS, or if the preclinical studies would have been done in a KI, transgenic or pff model of DLB.

Response: Again, the objective of our report is to demonstrate an effect of neflamapimod on basal forebrain cholinergic degeneration and dysfunction in the most relevant preclinical and clinical contexts, i.e., in a DS mouse model and DLB, respectively. We agree that the DS (Ts2) mouse model is not a disease model for DLB. However, both the Ts2 mouse model and patients with DLB demonstrate Rab5+ endosomal pathology and a similar pattern of cholinergic degeneration in the basal forebrain. Further, given the comment from the reviewer and at the request of the editor, we have added the following to the Discussion to make a case for similar pathogenic mechanisms underlying the cholinergic degeneration in the two contexts:

“As the great majority of individuals with DS develop early onset AD (EOAD) and there is increased APP expression in both human DS and in the DS mouse, the Ts2 mouse model is regarded as a model for EOAD. While our clinical trial was conducted in patients with DLB, EOAD and DLB both have major pathology in the basal forebrain cholinergic system and have similar cortical atrophy patterns (references 55, 57, 63-65). Furthermore, α -synuclein impairs retrograde axonal transport and BDNF signaling, in association with Rab5 and Rab7 accumulation (reference 66) and reducing endogenous α -synuclein in an APP transgenic mouse decreases Rab5 protein levels and prevents degeneration of cholinergic neurons (reference 67). Conversely, in mouse models of DLB, amyloid beta (A β) plaques promote seeding and spreading of α -synuclein (reference 68) and immunotherapy against A β and α -synuclein were additive against cholinergic fiber loss (reference 69). This literature, combined with our findings, suggests that, though the initiating factors may be different in EOAD and DLB, there is a common pathogenic path to cholinergic degeneration involving Rab5”

In addition, there is no data showing the mechanistic connection between Rab5 and p38 alpha that is primarily a glial cell kinase, while most of the endosomal pathology with Rab5 in human brains and AD models is neuronal. The authors have not shown convincingly that there is target engagement of neflamapimod in the CNS, this is a compound over 500 daltons with poor Lipinsky rule profile that have low BBB penetration. The other problem with neflamapimod is that it also inhibits other p38 isoforms and that this compound has failed in AD clinical trials, for these reasons more selective compounds with better CNS profile have been developed by some labs including the Van Eldik group.

Response: That there is a mechanistic link between Rab5 and p38 is long established in the scientific literature [starting with reference #21, Cavalli, V., *et al.* The stress-induced MAP kinase

p38 regulates endocytic trafficking via the GDI:Rab5 complex. *Mol Cell* **7**, 421-432 (2001); also reviewed in *Int. J. Mol. Sci.* 2020, **21**, 5485; doi:10.3390/ijms21155485]. In our report, we then take that established connection and evaluate the ability of neflamapimod to reduce Rab5 activity and in doing so reduce Rab5-mediated pathogenic effects on cholinergic neurons, both of which actions we demonstrated in the Ts2 mouse model. Moreover, target engagement was clearly demonstrated in the Ts2 mouse model, as along with reduced Rab5 activity, we showed reduced p38 and phospho-p38 expression, as well a significant reduction in downstream targets of p38, namely MK2 and MNK1. With respect to the DLB clinical study, there is now direct means to measure the above markers of target engagement, as they are all intraneuronal markers. However, as discussed in our report, the best clinical biomarkers of engaging cholinergic neurons are cognitive tests of attention, and potentially the Timed Up and Go test, both of which drug effects vs. placebo were demonstrated. In previous clinical studies effects of neflamapimod were demonstrated on CSF levels of IL-8 in a clinical pharmacology study (discussed reference #24), as well in a phase 2b study in AD on CSF levels of ptau and total tau, with significant reductions vs. placebo at week 24.

Regarding p38 being a “glial kinase” it is correct that in the healthy brain, p38 α is predominantly expressed in microglia and astrocytes and only at low levels in neurons. However, the expression and activation state are increased under conditions of cellular stress and disease (reference #27; we also showed that the expression of phospho-p38 α is increased in Ts2 mice relative to wild-type mice, fig.2d). Further, more recent studies in animal disease models with the gene for the alpha isoform (MAPK14) knocked out specifically in neurons (references #28, 48, 49) or using specific p38 α inhibitors (reference #22), argue that neuronal p38 α is also a relevant therapeutic target reference #58). The current work the provides additional support for pharmacological effects mediated through targeting neuronal p38 α . While we cannot be certain that the effects are not mediated through downstream effects of reducing glial p38 activity, we favor a neuronal effect as we have demonstrated (1) intra-neuronal biological effects, (2) it is more direct and economical, and (3), as discussed in the Discussion, the plasma drug concentrations achieved in the current clinical study are lower than that affecting cytokine production but consistent with that required for a neuronal effect (references 23, 24).

Regarding the neflamapimod-related comments, *i.e.*, low blood-brain-barrier (BBB) and inhibition of other p38 isoforms, they are simply not correct. To begin with, the compound has a molecular weight of 436 (not “over 500”). More importantly, regardless of the predicted BBB penetration, the measured brain concentrations in multiple pre-clinical species and in the CSF of humans indicates neflamapimod is robustly BBB penetrant, with brain concentrations in humans estimated to be two-fold higher than in plasma (reference 24, and EIP Pharma data on file). One reason for the excellent BBB penetration is thought to be the two fluorine molecules on the compound, which were placed there by the medicinal chemists to prevent metabolism. In doing so, the medical chemists inadvertently may have obtained BBB penetration, as fluorine molecules are known to increase BBB penetration (see “Fluorinated molecules as drugs and imaging agents in the CNS” *Top Med Chem* 2006;6(14):1457-64. doi: 10.2174/156802606777951046”).

Regarding isoform selectivity, the literature documented data from the originator company, Vertex Pharmaceuticals, demonstrates 20-fold selectivity against the beta isoform and no activity against delta and gamma isoforms (Duffy et al, "The Discovery of VX-745: A Novel and Selective p38 α Kinase Inhibitor" ACS Med. Chem. Lett. 2011, 2, 758–763, [dx.doi.org/10.1021/ml2001455](https://doi.org/10.1021/ml2001455) . That result has been independently confirmed in studies by the leading academic group profiling kinases, studies in which there was a 25-fold selectivity for p38 α against p38 beta, and no binding of the other isoforms was seen (see supplemental information in Davis et al, Comprehensive analysis of kinase inhibitor selectivity, Nature Biotech, **29**, pages 1046–1051 (2011).

In any case, the compound specific comments from the Reviewer would have been more relevant for neflamapimod at an earlier stage of development (*i.e.*, before entry into the clinic). At this time, the clinical data, in particular the ones in the current work, provide a better assessment of the potential of the compound from a pharmacological and clinical efficacy standpoint to act in the CNS; and the greater than 300 patients' clinical safety data, including data at substantially higher doses than the top dose of 40mg TID dose level in the DLB clinical study, has demonstrated excellent tolerability.